# Waste milk humification product can be used as a slow release nano-fertilizer

Yanping Zhu[1], Yuxuan Cao[1], Bingbing Fu[1], Chengjin Wang [2], Shihu Shu[1], Pengjin Zhu[3], Dongfang Wang[1], He Xu[1], Naiqin Zhong[4] & Dongqing Cai [1]✉

The demand for milk has increased globally, accompanied by an increase in waste milk. Here, we provide an artificial humification technology to recycle waste milk into an agricultural nano-fertilizer. We use KOH-activated persulfate to convert waste milk into fulvic-like acid and humic-like acid. We mix the product with attapulgite to obtain a slow-release nano fulvic-like acid fertilizer. We apply this nano-fertilizer to chickweeds growing in pots, resulting in improved yield and root elongation. These results indicate that waste milk could be recycled for agricultural purposes, however, this nano-fertilizer needs to be tested further in field experiments.

Over the past several decades, the growth in the global demand for milk has been accompanied by an increase in the disposal of waste milk (WM)[1]. Of the 600 million tons of raw milk produced on a global scale, 100 million tons (approximately 13%) were wasted in 2009[2], and the amount showed an increasing trend. By far, the majority of waste milk was disposed directly to receiving water bodies, inducing water eutrophication and threatening aquatic life[3]. On the other hand, waste milk was released into sewage system and discharged after conventional wastewater treatment, which may cause greenhouse gas emission and resource loss[4]. Considering that milk is typically rich in carbohydrates (4–5%), proteins (2.9–3.5%), and lipids (3.8–5.5%) with rare toxic compounds[5], using waste milk as liquid fertilizer or soil amendment may be a feasible recycling technique[6,7]. However, those perishable organic components tended to ferment in soil and thus lower crops germination rate, limiting direct use of waste milk in farmland[8,9]. Accordingly, it is hypothesized that the humification of waste milk may be an attractive route to realize the stabilization and nutrification of organics.

Artificial humification (i.e. artificially transforming biomaterials into humic substance instead of extracting from coal-related resources) can be achieved through microbial fermentation and abiotic physiochemical conversion techniques[10]. The whole humification process included initial decomposition and latter polymerization, along with the conversion of perishable organics into artificial humic-like acid (HLA) and/or fulvic-like acid (FLA)[11,12]. To be noted, the as-prepared HLA and FLA generally existed either in protonated or salt forms. Humic and fulvic-like acids are active components in organic fertilizer, both of which play key roles in promoting soil fertility and crop productivity[13]. Therein, fulvic-like acid possesses smaller molecular weights, higher solubility, and better bioactivity[14]. The microbial-mediated humification usually needs several weeks with relatively low HLA/FLA yield[8,15]. Comparatively, abiotic methods such as hydrothermal treatment and catalytic oxidation can shorten the duration to several hours and promote the conversion of organic precursors to humic and/or fulvic-like acids[16–18]. However, the present abiotic technologies always operate at elevated temperature (180-250 °C) and pressure (5–250 bar)[19,20], resulting in a high energy consumption and cost. Consequently, it is imperative to develop a heat-free alternative approach.

Base-activated persulfate (PS) is an efficient advanced oxidation process for the abatement of trace organic contaminants (e.g. pesticides, pharmaceutical and personal products) in water due to the production of highly reactive oxidative species (e.g. ·OH and $SO_4^{-}$•)[21,22]. Persulfate can also be used as one kind of radical initiators for polymerization and separation of organic pollutants[23,24]. Considering the critical role of polymerization in humification, PS-initiated radical polymerization reaction may be potentially used in the humification of organics in highly concentrated organic wastewater, such as waste milk[25–27]. Therein, reactive oxidative species may not only contribute to the initial degradation of macromolecules into small organic

[1]College of Environmental Science and Engineering, Donghua University, Shanghai 201620, People's Republic of China. [2]Department of Civil Engineering, University of Manitoba, Winnipeg, MB R3T 5V6, Canada. [3]Guangxi Subtropical Crops Research Institute, Nanning 530000, People's Republic of China. [4]Institute of Microbiology, Chinese Academy of Sciences, 100101 Beijing, People's Republic of China. ✉e-mail: dqcai@dhu.edu.cn

compounds, but also produce multiple reactive radical intermediates which can function as building blocks to build up humic and/or fulvic-like acids via radical-induced chain reactions[12,27–29]. Additionally, the radical-induced polymerization was typically exothermic and thus humification may occur under heat-free conditions[30]. Importantly, the conversion of waste milk into HLA/FLA-rich fertilizer may potentially maximize the nutrition value of organic components compared with conventional degradation or separation treatment.

In this work, KOH-activated PS (KOH/PS) is used to explore the humification of waste milk. The derived humic and fulvic-like acids in the humification product are characterized and quantified. For the convenience and cost-effectiveness of package, transportation, storage and optimization of utilization efficiency, the granulation process is employed to fabricate a spheric slow-release nano fulvic-like acid fertilizer via mixing the liquid product with a typical nanoclay attapulgite (ATP, $(Mg, Al)_4(Si)_8(O, OH, H_2O)_{26} \cdot nH_2O$, with mean diameter of approximately 30 nm). The release behavior and promotion effect on plant growth are evaluated. Additionally, radical identification and material characterization are conducted to explore the involved humification and slow-release mechanisms. This work highlights a

rapid heat-free artificial humification technology for the recycling of waste milk in sustainable agriculture.

## Results

### Humification of waste milk by KOH/PS and condition optimization

Three-dimensional Excitation-Emission matrix fluorescence (3D-EEM spectrum, Fig. 1) was used to determine the changes in composition of waste milk during treatment process. Generally, fluorescence spectra were divided into five regions: regions I (EX <250 nm, EM <330 nm) and II (EX <250 nm, EM 330–380 nm) referred to simple aromatic proteins, regions III (EX <250 nm, EMå 380 nm) and V (EXå 250 nm, EMå 380 nm) were related to fulvic and humic-like acids, region IV (EXå 250 nm, EM <380 nm) referred to soluble microbial by-product-like material (SMP). Meanwhile, fluorescence region integral (FRI) method was applied for the quantification of EEM results (Table S1) according to the fluorescence intensity and percentage fluorescence response of each region. As shown in Fig. 1i, the fluorescence of waste milk was mainly in regions II and IV, indicating proteins and SMP as its main components. After treatment of waste milk by PS alone (Fig. 1a), the fluorescence

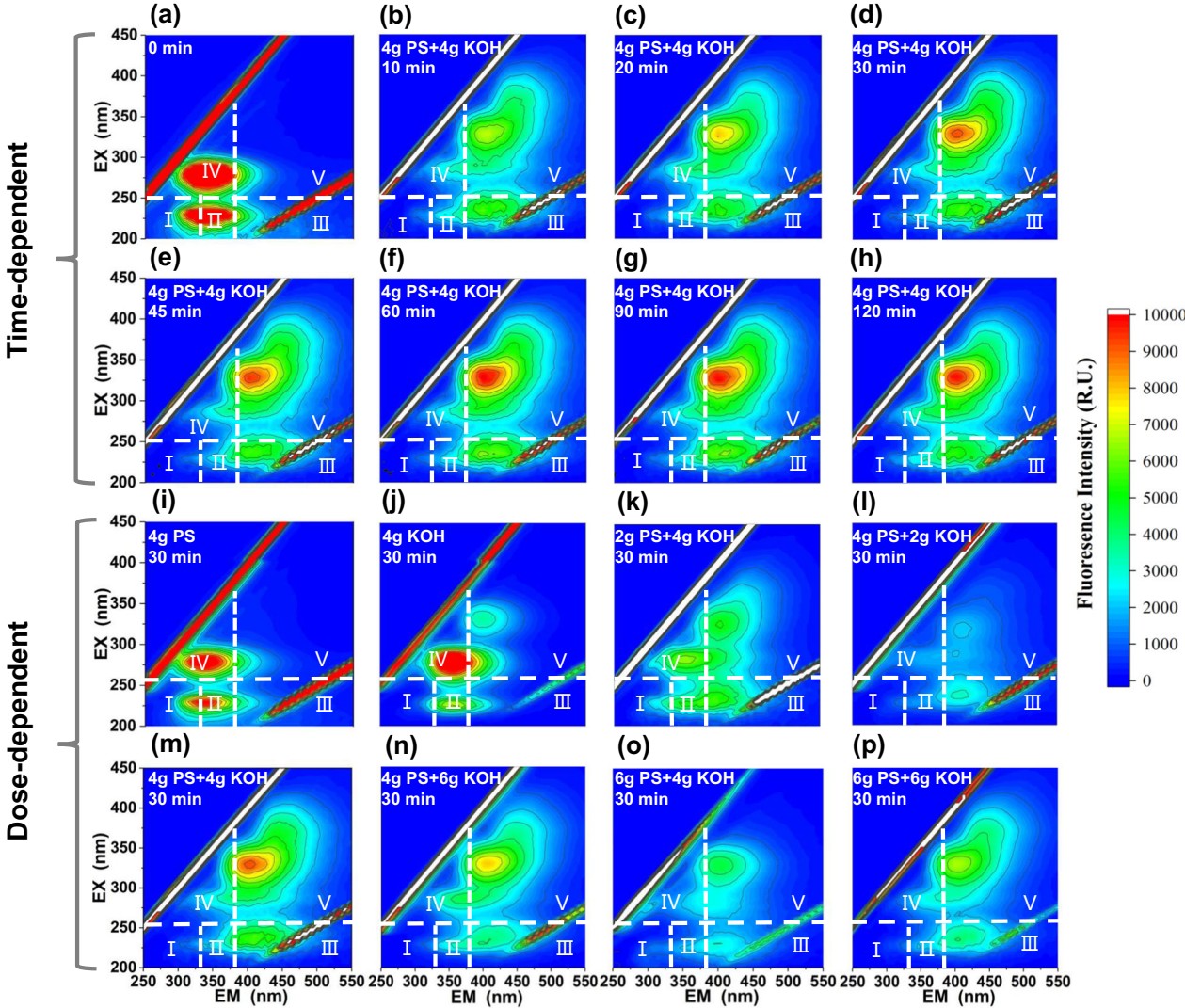

**Fig. 1 | Three-dimensional excitation–emission matrix fluorescence (3D-EEM) spectra of waste milk (WM) treated by KOH/persulfate (KOH/PS) under different conditions.** EEM spectra variation of waste milk treated by 4 g KOH/4 g PS at 0 min (**a**), 10 min (**b**), 20 min (**c**), 30 min (**d**), 45 min (**e**), 60 min (**f**), 90 min (**g**), and 120 min (**h**). EEM spectra of treated waste milk by 4 g PS (**i**), 4 g KOH (**j**), 2 g PS/4 g KOH (**k**), 4 g PS/2 g KOH (**l**), 4 g PS/4 g KOH (**m**), 4 g PS/6 g KOH (**n**), 6 g PS/4 g KOH (**o**) and 6 g PS/6 g KOH (**p**). Experimental conditions: waste milk volume = 50 mL, persulfate potassium (KOH) dosage = 0–6 g, ambient temperature = 22 ± 1°C, and reaction time = 0–2 h. Each experiment was performed three times independently with similar results. Source data are provided as a Source data file.

intensities in regions II and IV decreased, likely owing to direct PS oxidation of proteins and SMP. In addition, after being treated by KOH alone (Fig. 1b), the fluorescence intensities in regions II and IV decreased along with the appearance of slight fluorescence in region V, probably due to base-catalyzed hydrolysis and humification. Notably, the fluorescence further moved towards region III (fulvic-like acid) and V (humic-like acid) after the simultaneous addition of PS and KOH (Fig. 1c, d). Compared with waste milk (Fig. 1i), $P_{III}$ and $P_V$ of 4 g KOH/2 g PS-treated waste milk (Fig. 1c) significantly increased from 15.8% and 6.5% to 24.6% ($p = 8.6E−08$ by two-tailed Student's $t$ test) and 19.7% ($p = 5.6E−08$ by two-tailed Student's $t$ test), respectively, proving their synergistic effect on waste milk's humification probably according to Eqs. (1), (2)[31].

$$2S_2O_8^{2-} + 2H_2O \rightarrow 3SO_4^{2-} + SO_4^- \cdot + O_2^- + 4H^+ \tag{1}$$

$$SO_4^- \cdot + OH^- \rightarrow SO_4^{2-} + \cdot OH \tag{2}$$

$$2SO_4^- \cdot \rightarrow S_2O_8^{2-} \tag{3}$$

$$2 \cdot OH \rightarrow H_2O + 1/2 O_2 \tag{4}$$

In order to maximize the degree of waste milk humification, the dosages of PS and KOH were optimized in the range of 2–6 g (Fig. 1c–h and Fig. S2). With the increase of their dosages, the fluorescence intensities (Fig. 1c–h) and percentage fluorescence responses of regions III and V (Table S1) increased initially (from 2 to 4 g) and then decreased (from 4 to 6 g), achieving the maximum at 4 g PS and 4 g KOH. Therein, the decrease trend at overdoses (above 4 g) of PS and KOH may be due to self-quenching of radicals (Eqs. (3)–(4))[32,33] or enhanced decomposition of the precursors/humification product[34,35]. Besides, the reaction time was optimized with 4 g PS and 4 g KOH. As shown in Fig. 1i-p, the fluorescence intensity increased with time and became stable after 60 min, which was consistent with the results in Table S1. Therefore, the optimal humification conditions were chosen at dosages of 4 g PS/4 g KOH and reaction time of 60 min, showing the $P_{III}$ and $P_V$ of 29.5 and 32.3%, respectively. This result could be also supported by the UV-vis results in Fig. S1.

Additionally, weight method (introduced in Methods section) was used to precisely quantify humic-like acid, fulvic-like acid, and soluble humic acid (SHA, a mixture of HLA and FLA) in the product under optimal condition. Contents of humic-like acid and fulvic-like acid likely in protonated or potassium salt forms were calculated to be 44.4(±0.2)% and 25.5(±0.4)% respectively according to Eqs. (6)–(9) (Table S2). Importantly, the contents in the product of this work were much higher than the reported humic-like acid (2.6–8%) and fulvic-like acid (0.7–4%) results via composting of food wastes[36–38]. On the other hand, Table S3 showed that the organic carbon loss was 20.5% during the whole process, which was relatively low compared with composting of food waste (30–60%)[39]. That is, compared with composting, this technology could potentially conserve more organic carbon in the product.

## Characterization of HLA/FLA in product

To investigate the time-dependent structural variation of treated waste milk during the humification, Fourier transform infrared spectrometer (FTIR) spectra were analyzed at different intervals (0, 2, 5, 15, 30, and 60 min). As shown in Fig. 2a, the characteristic peaks of C-N (1400 cm⁻¹), N-H (620 and 1563 cm⁻¹) and aromatic -OH (1100 cm⁻¹)[40] became stronger within initial 30 min and then (from 30 to 60 min) stable. While the peaks of aliphatic -OH (3200–3500 cm⁻¹) and aldehyde C = O (1750 cm⁻¹) became weakened similarly. These results likely

indicated a process of amine aldehyde condensation, dehydration and cyclization in Maillard reaction during the humification[41].

Furthermore, the groups of the freeze-dried product and extracted fulvic-like acid were compared with fulvic acid (FA) standard. The peaks of carboxyl -OH (2923 and 2850 cm⁻¹), aromatic -OH (1100 cm⁻¹), C-N (1400 cm⁻¹), and N-H (1563 cm⁻¹) were consistently found in fulvic acid standard and product, revealing the successful synthesis of FLA[42]. Their higher peaks of product demonstrated the greater amounts of these groups in product than those of FA standard, which was similarly observed in previous artificial humification products[20]. Besides, the extracted fulvic-like acid similarly exhibited obvious peaks of aromatic OH (1100 cm⁻¹) and aliphatic OH (3200–3500 cm⁻¹), evidencing the similarity of extracted fulvic-like acid and fulvic acid standard.

The solid-state ¹³C nuclear magnetic resonance (¹³C NMR) spectra in Fig. 2B could be generally divided into four chemical regions including aliphatic carbons (0–50 ppm), oxygenated aliphatic carbons (50–100 ppm), aromatic carbons (100–160 ppm), and carboxyl/carbonyl carbons (160–220 ppm)[43]. As observed in the aliphatic carbon region, both waste milk and product had a prominent peak at 33 ppm due to the presence of aliphatic -CH in methyl, methylene, and methine groups. The higher peak intensity in product compared to waste milk represented the higher abundance of alkyl groups[44,45]. In aromatic carbon region, the peak at 106 ppm of waste milk disappeared along with emergence of a peak at 130 ppm. These results were likely due to the radical oxidation-induced ring opening and subsequent fused-ring formation. In the carboxylic/carbonyl carbon region, the peaks around 170 ppm of product were much higher than those of waste milk, confirming the generation of more carboxylic groups probably due to carboxylation reaction. The typical peaks of alkyl (around 35 ppm), aromatic (130 ppm), and carboxylic (170 ppm) groups in product could be also found in spectrum of fulvic acid standard, suggesting the formation of fulvic-like acid.

Higher resolution X-ray photoelectron spectroscopy (XPS) was conducted to further investigate the composition variation during humification. Figure 2C, D showed that C1s peak can be deconvoluted into peaks located at 284.5, 285.5, 285.9 and 287.8 eV, corresponding to C-C, C-O, C=O and O-C=O[46]. Compared with waste milk, the C=O peak of product became weakened, while those of C-O and O-C=O were intensified, which fully agree with the results from FTIR and NMR analysis, probably indicating that hydroxylation and carboxylation reaction occurred. In Fig. S3A, B, N1s peak of product was divided into 399.4, 400.1 and 403.2 eV, assigned to amide (pyrimidine/peptide N), amide/pyrrole, and primary amine/protonated amine respectively[47]. Compared to waste milk, there was an increase in amide/pyrrole and presence of primary amine/protonated amine in product, which may be resulted from the Maillard reaction.

Overall, the chemical structure of product contained more active groups (-COOH, -CNOH, -OH) than fulvic acid standard. These active groups were reported to contribute to humic-like acid/fulvic-like acid's function in exerting hormone-like effect in plant or improving the retention capacity of soil water and nutrient[48]. During the treatment, multiple reactions including Maillard, hydroxylation, carboxylation, and ring opening/rebuilding may proceed among carbohydrates, proteins, and fat, contributing to waste milk's humification and fulvic-like acid formation. Therein, Maillard reaction probably played a key role in the process, which could be further verified by the color change, EEM and UV analyses (Note S3).

## Role of radicals in humification

In order to explore the contribution of different radicals in the humification of waste milk by KOH/PS, both electron paramagnetic resonance (EPR) measurement and quenching experiments were performed. EPR results in Fig. 3A, B showed that •OH and $SO_4^- \cdot$ were identified at 2 and 30 min[49]. Meanwhile, PS concentration decreased sharply to 58.5% within 2 min and then slowly to 15.6% at 30 min (Fig.

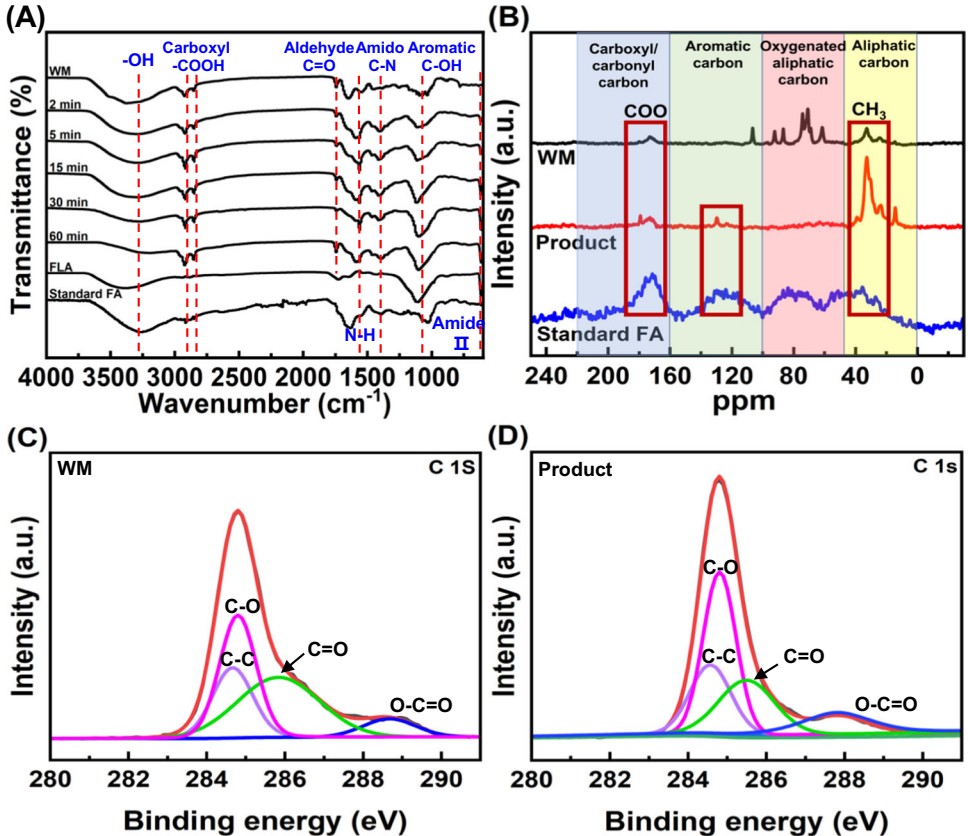

**Fig. 2 | Structural characterization of waste milk humification product.**
**A** Fourier transform infrared spectrometer (FTIR) spectra of standard fulvic acid (FA) and treated waste milk (WM) at different intervals (0, 2, 5, 15, 30, and 60 min). **B** Solid-state $^{13}C$ nuclear magnetic resonance ($^{13}C$-NMR) spectra of waste milk, product and standard FA. **C, D** X-ray photoelectron spectroscopy (XPS) spectra of C1s for waste milk and product. Humification conditions: [persulfate potassium]$_0$ = 80 g/L, [KOH]$_0$ = 80 g/L, ambient temperature=22 ± 1 °C, reaction time = 1 h. Each experiment was performed three times independently with similar results. Source data are provided as a Source Data file.

S4). Therefore, the generation of •OH and $SO_4^-$• was likely due to the decomposition of PS under alkaline activation (Eqs. (1)–(2)). For quenching tests, *tert*-butanol (TBA) was used as a quencher for •OH ($k_{•OH/TBA} = (3.8–7.6) \times 10^8 M^{-1} s^{-1}$), while ethanol anhydrous (EtOH) for both •OH and $SO_4^-$• ($k_{•OH/EtOH} = 1.6 \times 10^7 M^{-1} s^{-1}$, $k_{SO_4^-•/EtOH} = 1.9 \times 10^9 M^{-1} s^{-1}$)[50]. To be noted, the obvious shifts of florescent components in regions II and IV during humification (Fig. 1m) indicated that those components were key precursors involved in the humification, while the addition of *tert*-butanol or ethanol anhydrous to waste milk (without PS/KOH) rarely changed the florescence in regions II and IV (Fig. S5), indicating that *tert*-butanol or ethanol anhydrous did not interfere with those humification precursors greatly. As shown in Fig. 3C, the addition of *tert*-butanol or ethanol anhydrous in the humification system led to minor change or shift of EEM peaks in regions II and IV relative to waste milk, indicating a suppressed degradation or transformation of proteins and amino acids due to the quenching of •OH or $SO_4^-$•. This was reasonable considering previously reported high reaction rates ($10^7–10^9 M^{-1} s^{-1}$) between radicals and amino acids[51,52]. Therefore, •OH generated through the proposed multiple reactions (Eqs. (1)–(4)) may play a key role in the humification of waste milk, which was consistent with the reported oxidation (e.g. hydroxylation and carboxylation)[52] and polymerization (e.g. Maillard reaction) effects of •OH[53,54].

Interestingly, during the KOH/PS treatment, the reaction temperature was observed with a significant increase (up to 61 °C) within the initial 10 min in waste milk system (Fig. 3D), while only a minor rise occurred in water system. This phenomenon reflected that a lot of heat was generated during the reaction between KOH/PS and organics in waste milk, which would further activate PS to accelerate the

production of radicals[55]. This is consistent with the heat release in natural composting process, although the heat release occurs over a much longer time if we rely on the microorganisms to do the work. Then, the temperature in waste milk system gradually decreased to 21 °C within the following 50 min, which was relevant to the large consumption of PS (68.4%) within the first 10 min. As for the heat intensively released during 60 min humification, that would be easily gathered and recycled for PS activation in industrial application, thus potentially saving the dosage of KOH for activation.

Gel permeation chromatography (GPC) result in Fig. 3E showed that the molecular weight of filtered waste milk was in range of 500–50,000 Da with the most distribution around 1000 Da. After treatment, the molecular weight of filtered product was reduced to about 200–50,000 Da with the most distribution around 500 Da. This finding was probably related to the degradation of macromolecules into small ones, followed by polymerization via radical reactions. It seemed that the polymerization process did not generate molecules larger than the species originally existed in waste milk. The most distribution around 500 Da, which coincided with the reported molecular weight (397-1203 Da) of fulvic acid molecular model structure, likely proved the high contents of FLA in product[56]. Generally, fulvic-like acid with lower molecular weights were more easily assimilated by plant[57], therefore the as-prepared product herein probably possessed a good promotion effect on plant growth.

## Fabrication and characterization of slow-release nano fulvic-like acid fertilizer
For the application convenience and enhancement of HLA/FLA utilization efficiency, the humification product was mixed with attapulgite

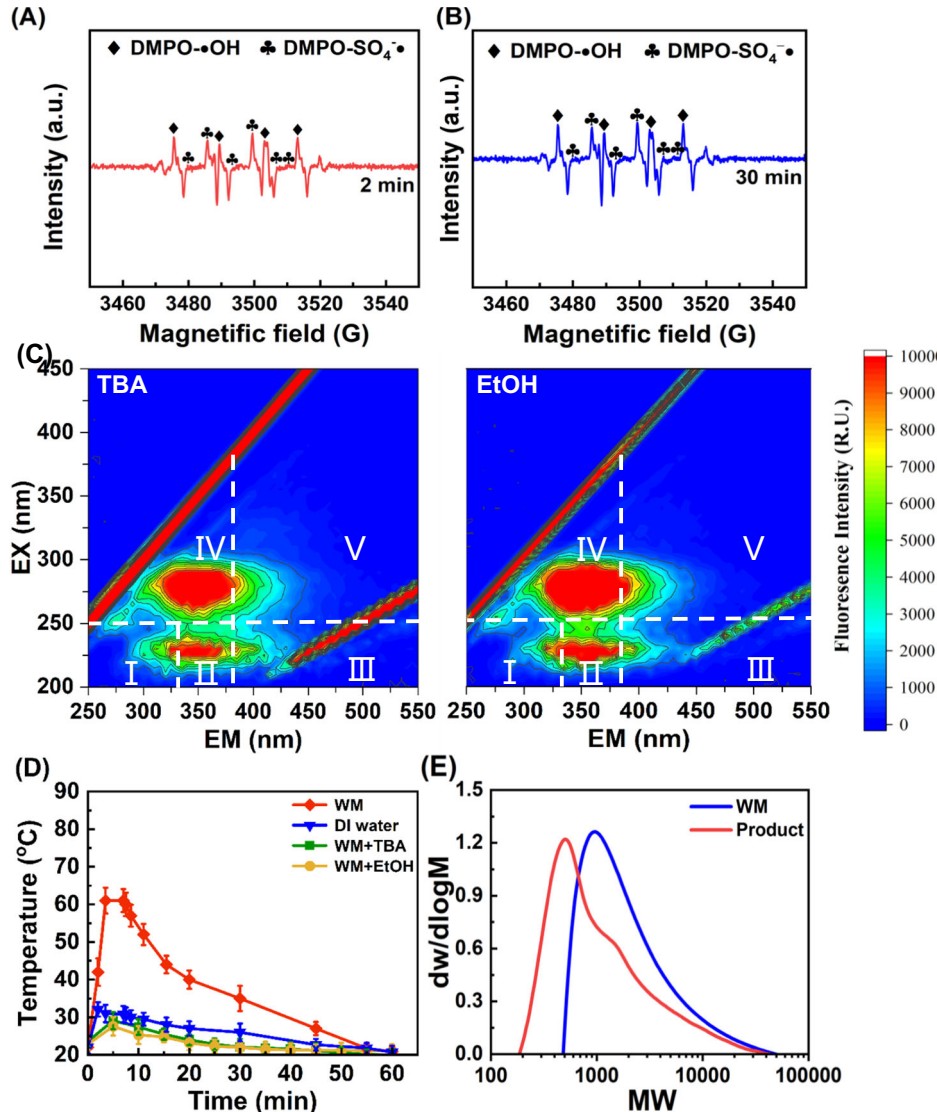

**Fig. 3 | Evidence of radical-mediated polymerization during humification.**
Electron paramagnetic resonance (EPR) spectra of (**A, B**) •OH and $SO_4^-$• during the treatment of waste milk (WM) by KOH/persulfate (KOH/PS) under optimum conditions. **C** Three-dimensional Excitation-Emission matrix fluorescence (3D-EEM) spectra of KOH/PS-treated waste milk in the presence of *tert*-butanol (TBA) and ethanol anhydrous (EtOH). **D** Temperature variation in different systems. Error bars represent the standard deviations from triplicate tests (*n* = 3). **E** Gel permeation chromatography (GPC) spectra of filtered waste milk and product. Experimental conditions: [persulfate potassium]$_0$ = 80 g/L, [KOH]$_0$ = 80 g/L, [DMPO]$_0$ = 200 mmol/L, [TBA]$_0$ = 10 mol/L, and [EtOH]$_0$ = 10 mol/L, ambient temperature = 22 ± 1 °C, reaction time = 1 h. Experiments in (**A**–**C**, **E**) were performed three times independently with similar results. Source data are provided as a Source Data file.

and xanthan gum to prepare slow-release nano fulvic-like acid fertilizer with diameter of approximately 3 mm. Therein, attapulgite was a typical nanoclay consisting of nanorods with mean diameter of approximately 30 nm and widely used as a slow-release nanocarrier[58]. As illustrated in Fig. 4A–D, the freeze-dried waste milk showed a smooth surface and became rough after treatment by KOH/PS. Attapulgite possessed a porous nanonetworks structure formed by numerous micro-nano rods, which was favorable for the loading of humic/fulvic-like acids. In slow-release nano fulvic-like acid fertilizer, the products distributed in the pores and on the rods' surface, indicating the successful adsorption of humic/fulvic-like acids in micro-nano pores of attapulgite. The formed nanonetworks of slow-release nano fulvic-like acid fertilizer may facilitate the slow-release of humic/fulvic-like acids.

As shown in Fig. 4E, the FTIR characteristic peaks of product (1563 cm⁻¹ for -NH and 1410 cm⁻¹ for -CN) and attapulgite (3617 and 3555 cm⁻¹ for -OH, 988 cm⁻¹ for in-layer Si−O−Si) could be found in

the spectrum of slow-release nano fulvic-like acid fertilizer[59], implying the successful loading of product in attapulgite. Besides, the peak intensity at 988 cm⁻¹ in slow-release nano fulvic-like acid fertilizer became stronger compared with attapulgite alone. The N-H peak at 1648 cm⁻¹ of slow-release nano fulvic-like acid fertilizer left-shifted compared with product. These results probably indicated the formation of hydrogen bonds between Si−O−Si of attapulgite and N-H of humic/fulvic-like acids[58], which may be conducive to the slow-release of slow-release nano fulvic-like acid fertilizer. In addition, no obvious new peak appeared in the spectrum of slow-release nano fulvic-like acid fertilizer compared with attapulgite or product, implying that the loading of humic/fulvic-like acids is mainly through physical interactions during the fabrication process[58].

Figure 4F showed that the characteristic peaks of $K_2SO_4$ could be found in X-ray diffraction (XRD) spectrum of product according to standard atlas in JADE, demonstrating that $K_2SO_4$ was the main

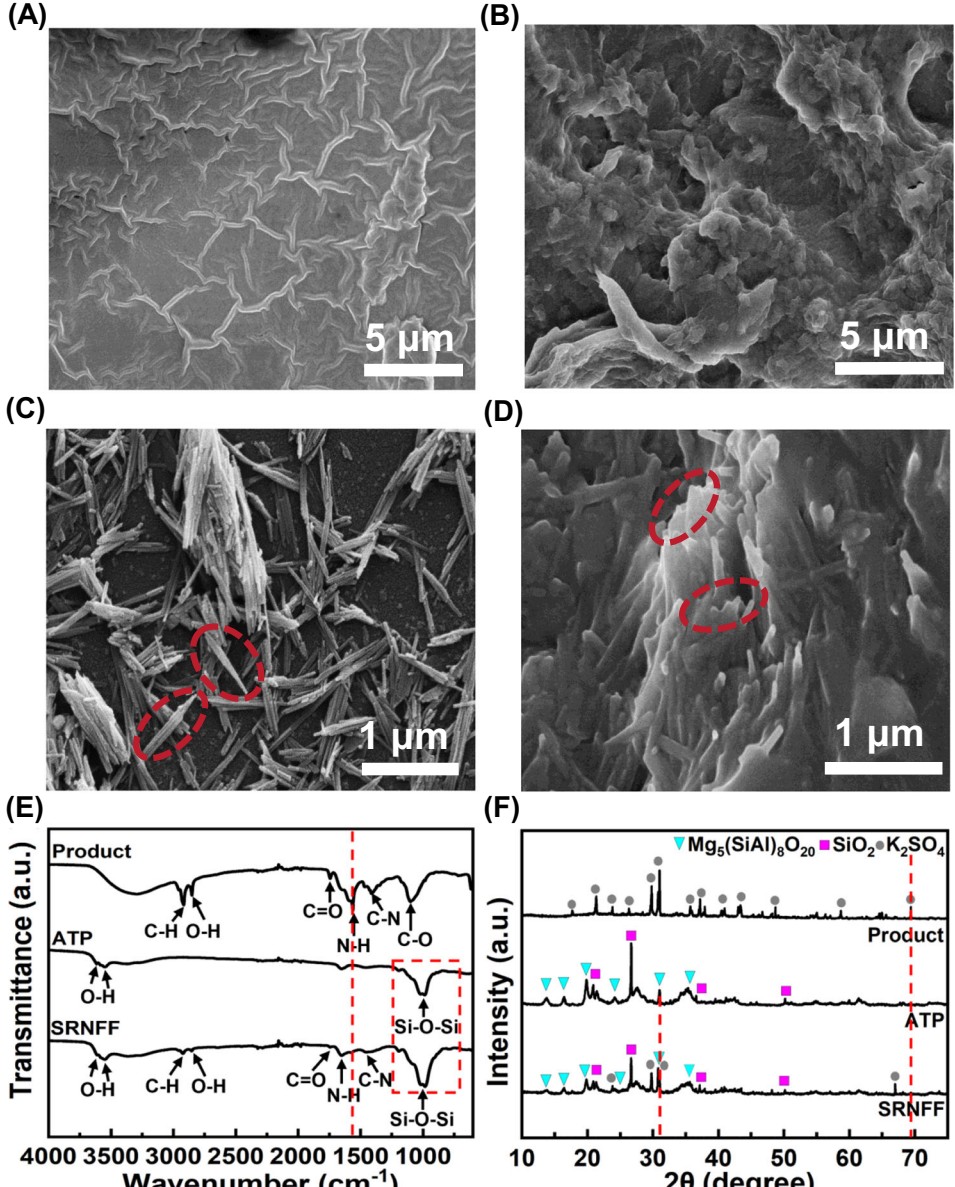

**Fig. 4 | Characterization of prepared slow-release nano fulvic-like acid fertilizer (SRNFF).** Scanning electron microscopy (SEM) images of (**A**) freeze-dried waste milk (WM), (**B**) product, (**C**) attapulgite (ATP) and (**D**) SRNFF. **E** Fourier transform infrared spectrometer (FTIR) spectra and (**F**) X-ray diffraction (XRD) patterns of product, ATP and SRNFF. Humification conditions: [persulfate potassium]$_0$ = 80 g/L, [KOH]$_0$ = 80 g/L, ambient temperature = 22 ± 1 °C, reaction time = 1 h. Each experiment was performed three times independently with similar results. Source data are provided as a Source Data file.

crystal component and humic/fulvic-like acids may possess amorphous structure. In other words, the product could be seemed as a kind of combined organic-inorganic fertilizer containing both $K^+$ and humic/fulvic-like acids[60]. The production of $K_2SO_4$ was likely attributed to the reaction between PS and KOH (Eqs. (1)–(2)). The peaks of $Mg_5(SiAl)_8O_{20}$, $SiO_2$ and $K_2SO_4$ could be observed in spectrum of slow-release nano fulvic-like acid fertilizer, also confirming the loading of product in attapulgite. Notably, compared to the product, the peak at 2θ = 69° of $K_2SO_4$ in slow-release nano fulvic-like acid fertilizer left-shifted, probably because attapulgite changed the crystal orientation of $K_2SO_4$. Also, the peak at 31° in slow-release nano fulvic-like acid fertilizer left-shifted compared to attapulgite, probably due to the intercalation of humic/fulvic-like acids or $K_2SO_4$ into attapulgite crystal layers, which was beneficial to the slow-release of humic/fulvic-like acids from slow-release nano fulvic-like acid fertilizer.

## Slow-release behavior of slow-release nano fulvic-like acid fertilizer

Considering that the alkaline property of the product may increase the pH of acidic soil, the humic/fulvic-like acids release behavior of slow-release nano fulvic-like acid fertilizer was investigated in aqueous solution at pH 3 to compare with pH 7 (the typical soil pH). The loadings of soluble humic acid and fulvic-like acid in slow-release nano fulvic-like acid fertilizer were determined to be 103.3 and 72.9 mg/g. The cumulative release concentration (RC) was calculated from the standard curves of soluble humic acid and fulvic-like acid at different pHs (Fig. S6). The release ratio (RR) was calculated according to Eq. (5). As shown in Fig. 5A, both soluble humic acid and fulvic-like acid were released slowly at pH 7 and close to be equilibrium in 96 h. The maximum cumulative release concentrations of soluble humic acid and fulvic-like acid were 1748 and 1475 mg/L, respectively, with corresponding release ratios of 56.4%

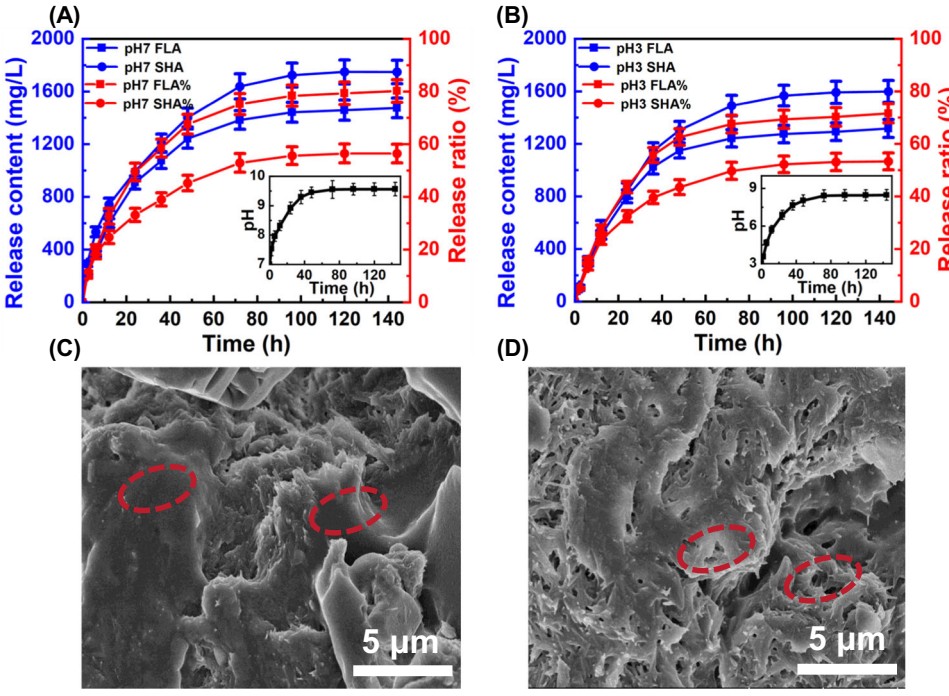

**Fig. 5 | Slow-release performance of slow-release nano FLA fertilizer (SRNFF).** **A**, **B** Cumulative released concentration and release ratio of fulvic-like acid (FLA) and soluble humic acid (SHA) from SRNFF at initial pHs of 7 and 3. Error bars in (**A**) and (**B**) represent the standard deviations from triplicate tests ($n = 3$). Scanning electron microscopy (SEM) images of SRNFF surface (**C**) before and (**D**) after release. Inserts in (**A**) and (**B**) represent pH variation in the aqueous slow-release systems. Experimental conditions: $[SRNFF]_0 = 30$ g/L and ambient temperature $= 22 \pm 1$ °C. Experiments in (**C**, **D**) were performed three times independently with similar results. Source data are provided as a Source Data file.

and 80.2%. Figure 5B showed the same release trend at pH 3, with a slightly slower release (close to reach equilibrium at 120 h) and lower release concentrations (1599 mg SHA/L and 1317 mg FLA/L), likely due to the lower solubility of humic-like acid at acidic conditions. To be noted, the majority of the released soluble humic acid was fulvic-like acid, accounting for 80.2 (pH 7) and 71.6% (pH 3) at equilibrium, which was consistent with the better solubility of fulvic-like acid compared with humic-like acid[61].

Meanwhile, with the release of humic/fulvic-like acids, the pH of the aqueous solution increased gradually, finally stabilizing at 9.6 and 8.5, respectively, indicating the potential application of slow-release nano fulvic-like acid fertilizer in acidic soil improvement. In addition, the color of the solution became yellow gradually during the process (Fig. S7) and plenty of micro-pores appeared in slow-release nano fulvic-like acid fertilizer after release (Fig. 5C, D), confirming the slow-release of humic and fulvic-like acids. The kinetics analysis result indicated that fulvic-like acid release fitted best to the First-order release model ($R^2 > 0.99$) (Fig. S8 and Note S1), implying the slow-release mechanism was relevant to simple adsorption[62]. Such adsorption was likely owing to the aforementioned hydrogen bonds and intercalation among attapulgite and humic/fulvic-like acids in Fig. 4E, F.

$$RR(\%) = (RC \times \text{solution volume})/(\text{loading} \times \text{slow} - \text{release nanofulvic} - \text{like acidfertilizerdosage}) \times 100\%$$

$$(5)$$

## The fertilization effect of slow-release nano fulvic-like acid fertilizer

As shown in Fig. 6, the growth of chickweeds in 21 days by treatments of waste milk, attapulgite, $K_2SO_4$, soybean meal-composted organic fertilizer (COF), product, and slow-release nano fulvic-like acid fertilizer (SRNFF) was compared with the Control. Fresh weight, as the most important index, exhibited an order of SRNFF>Product>COF>

$K_2SO_4$ > ATP>Control>WM in Fig. 6C, which was in line with the photograph in Fig. 6A, B. Therein, the average fresh weights of all the chickweeds in one pot with slow-release nano fulvic-like acid fertilizer (2.19 g), Product (1.49 g), COF (1.24 g), $K_2SO_4$ (1.15 g), and ATP (1.13 g) increased by 109% ($p < 0.05$ by one-way ANOVA Tukey's t test), 41.9% ($p < 0.05$ by one-way ANOVA Tukey's t test), 18.5% ($p < 0.05$ by one-way ANOVA Tukey's t test), 9.4% ($p < 0.05$ by one-way ANOVA Tukey's t test), and 7.3% ($p < 0.05$ by one-way ANOVA Tukey's t test) compared with Control (1.05 g), while that of WM (0.99 g) decreased by 5.4% ($p < 0.05$ by one-way ANOVA Tukey's t test). The fertilization effect of product relative to WM indicated the effectiveness of the humification process in transforming waste milk to a kind of high-performance fertilizer. The superior fertilization performance of product compared to COF and $K_2SO_4$ indicated the contribution of humic/fulvic-like acids. Notably, the much better fertilization of slow-release nano fulvic-like acid fertilizer than product and attapulgite indicated that slow-release could further improve utilization efficiency of nutrients. Their dry weight showed a similar trend to the fresh weight in Fig. S10. Meanwhile, Fig. 6D showed that both slow-release nano fulvic-like acid fertilizer and Product groups had germination rates of 100%, significantly higher than other treatments, suggesting the promotion effect of humic/fulvic-like acids on germination. Similarly, treatments with slow-release nano fulvic-like acid fertilizer and product greatly stimulated the development of leaves (Fig. 6E) by 104% ($p < 0.05$ by one-way ANOVA Tukey's t test) and 42.1% ($p < 0.05$ by one-way ANOVA Tukey's t test) compared with Control. As shown in Fig. 6F, the average taproot length of the chickweeds with ATP (8.58 cm), product (7.98 cm), and slow-release nano fulvic-like acid fertilizer (10.11 cm) were 38.6% ($p < 0.05$ by one-way ANOVA Tukey's t test), 24.9% ($p < 0.05$ by one-way ANOVA Tukey's t test), and 58.2% ($p < 0.05$ by one-way ANOVA Tukey's t test) higher compared with the Control (6.39 cm), implying their promotion effects on root elongation[63,64]. Similar plant-stimulation effects have been previously observed in studies using paper mill effluent-extracted or Fenton-derived fulvic-like acids[65,66].

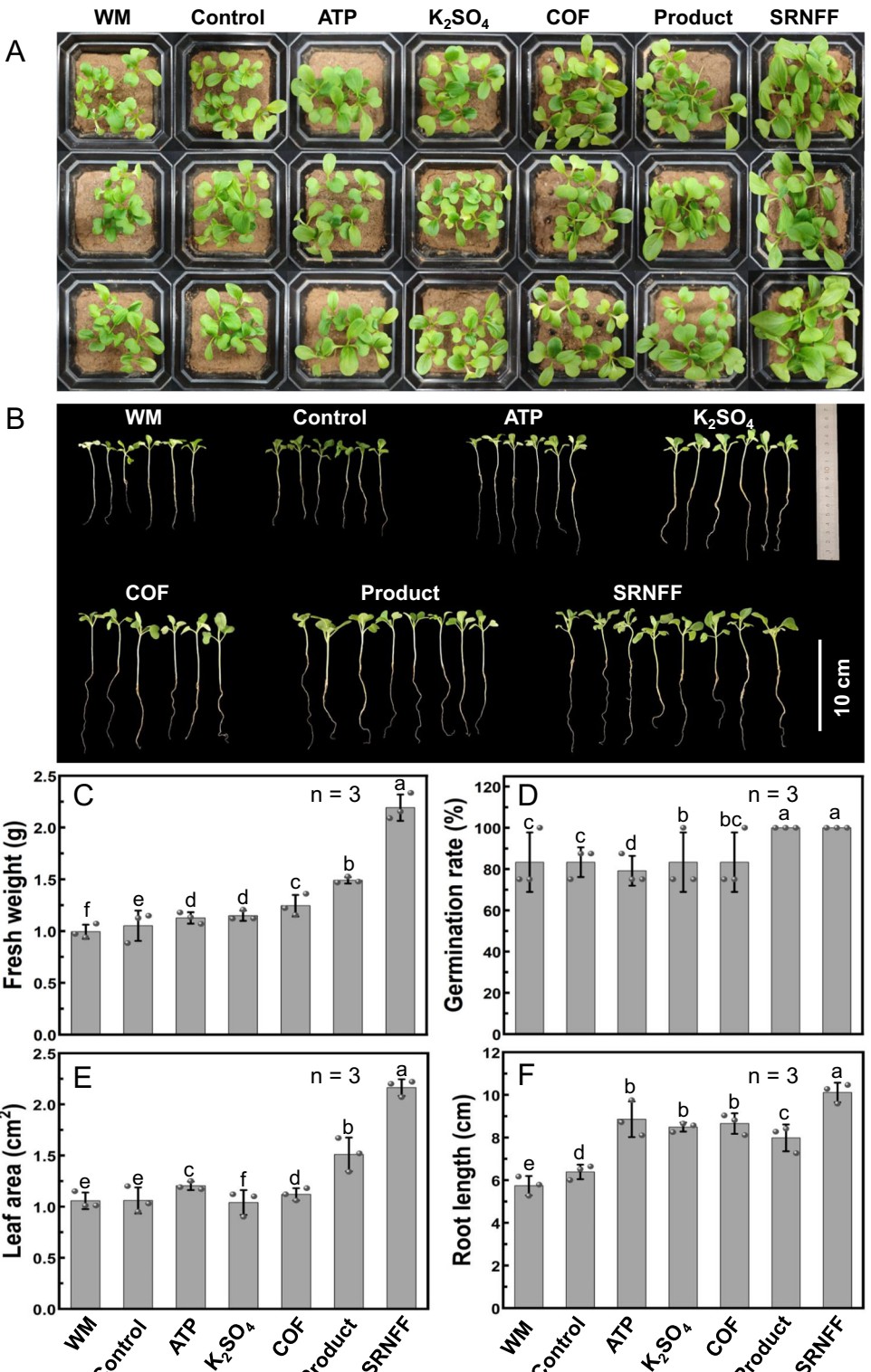

**Fig. 6 | Fertilization evaluation of the product and as-prepared slow-release nano fulvic-like acid fertilizer (SRNFF). A** Overall pot photograph, (**B**) growth photograph in representative pots, (**C**) fresh weight, (**D**) germination rate, (**E**) leaf area, and (**F**) taproot length of chickweeds in 21 d pot experiment with different treatments: WM, Control, ATP, $K_2SO_4$, COF, Product, and SRNFF refer to 400 g soil +1.4 mL waste milk (WM), 400 g soil alone, 400 g soil+0.7 g attapulgite (ATP), 400 g soil+0.14 g $K_2SO_4$, 400 g soil+1.5 g soybean meal-composted organic fertilizer (COF), 400 g soil+1.4 mL product, and 400 g soil+1.5 g SRNFF, respectively. Each pot contained 8 chickweed seeds. Each group was in triplicates and error bars represent the standard deviations. In each group, the fresh weight of the pot closest to the mean value was selected as the representative one in Fig. 6B. Comparison among results of different treatments were via one-way ANOVA analysis, and different letters refer to significant differences with Tukey's *t* test at $p < 0.05$ ($n = 3$).

**Table 1 | Operation cost comparison of this technology with conventional composting**

| Cost type (RMB/ton organic fertilizer) | This work (RMB) | Conventional composting (RMB) |
|---|---|---|
| Material | 150[a] | 80 |
| Labor | 20 | 50 |
| Electricity | 10 | 10 |
| Equipment depreciation | 20 | 40 |
| Maintenance | 4 | 10 |
| Plant depreciation | 2 | 20 |
| Package | 50 | 50 |
| Total | 256 | 260 |

[a]The material cost of the technology (150 RMB) was calculated based on the practical industry with KOH and PS dosages of both 15 kg in producing per ton solid organic fertilizer (meeting NY525-2021 organic fertilizer standard of China). Notably, the lower dosages compared with waste milk herein was probably because the temperature was observed to rise more easily in solid manure owing to the bulk effect, lower moisture, and different components.

Results of plant height and leaf chlorophyll content also proved the high fertilization effects of slow-release nano fulvic-like acid fertilizer and product (Fig. S10).

In addition, Table S4 showed the results of soil physicochemical indexes including pH, total N (TN), total P (TP), total K (TK), available P (AP) and available K (AK) in soil with different treatments. As for pH, slow-release nano fulvic-like acid fertilizer, product and attapulgite, with alkaline property, could increase the soil pH (initial pH 5.0) to 7.1, 7.3 and 7.4 respectively. Relatively, $K_2SO_4$ and COF showed lower effect on increasing soil pH. This result proved the effectiveness of slow-release nano fulvic-like acid fertilizer and product on acid soil amendment. As for the nutrients, the addition of slow-release nano fulvic-like acid fertilizer and product could increase soil nutrients especially available K from 96.12 to 237 ($p < 0.05$ by one-way ANOVA Tukey's $t$ test) and 232 mg/kg ($p < 0.05$ by one-way ANOVA Tukey's t test) respectively. Overall, the growth promotion and soil improvement effects of slow-release nano fulvic-like acid fertilizer were likely related to the slow-release of humic/fulvic-like acids, $K^+$, and $OH^-$ carried by attapulgite. The underlying structure-function relationship of the as-prepared slow-release nano fulvic-like acid fertilizer needs further investigation.

## Discussion

This paper highlighted a promising artificial humification technology for conversion of waste milk into products rich in fulvic-like acid (25.5%) and humic-like acid (18.9%) in 1 hour using KOH/PS under ambient temperature and pressure. This technology exhibited advantages of short duration and high fulvic-like acid yield, showing a potential to overcome the shortcomings of conventional composting. The results of EPR and quenching experiments indicated that KOH/PS could generate abundant reactive oxidative species such as •OH and $SO_4^-$•. The mechanism of waste milk humification may be related to radical-induced degradation and polymerization reactions, and the detailed mechanism needs to be explored in future. The results of product characterization revealed that fulvic-like acid had more active functional groups than natural fulvic acid, which may be relevant to reactions of hydroxylation, carboxylation and Maillard. Besides, results of pot experiment proved that the as-prepared slow-release nano fulvic-like acid fertilizer possessed growth-promoting and acidic soil-amending effects, indicating slow-release nano fulvic-like acid fertilizer's application potential in improving productivity of acidic soil (up to 3.95 billion hectares) worldwide[67].

This rapid abiotic humification method offered a "several birds with one stone" strategy to simultaneously solve recycling of waste milk, shortage of natural humic acid, and low fulvic-like acid content in traditional organic fertilizer. The technology has already been industrialized in several areas of China for humification of livestock manure. The cost of this technology was generally compared with conventional composting (Table 1) using pig manure as raw material. The cost difference mainly occurred in the treatment unit other than collection, transportation and utilization units. This technology could save operation cost by 4 RMB/ton because the saving costs (74 RMB/ton) of labor, plant/equipment depreciation and maintenance could cover that (70 RMB/ton) of materials. Additionally, a brief comparison of carbon loss and energy consumption/reuse was performed to assess the environmental impacts of this technology. Table S5 illustrated that this technology could reduce carbon loss by 62% according to the TOC loss (20.5%) during waste milk's humification[68]. Meanwhile, the intensively-generated heat (8.4 kJ/ton) was readily to collect for reuse. Overall, this efficient humification technology showed advantages of cost-effective and environmentally friendly. In the future, further life cycle cost (LCC) and life cycle assessment (LCA) will be performed for the comprehensive evaluation of its application potential. In addition, the product stability was evaluated, and no significant variation was found in its 3D-EEM fluorescence regions within 12 days (Fig. S11).

## Methods
### Chemicals
All chemical reagents were of analytical grade without further purification. KOH (purity≥95%), $K_2S_2O_8$ (PS, purity≥99.5%), hydrochloric acid (36.0-38.0%), *tert*-butanol (TBA, $C_4H_{10}O$, ≥99%), ethanol anhydrous (EtOH, $C_2H_6O$, 99.5%), potassium sulfate ($K_2SO_4$, ≥99%), potassium iodide (KI, 99%) and sodium bicarbonate ($NaHCO_3$, 99.5%) were purchased from Sinopharm Chemical Reagent Company (Shanghai, China). Standard samples of humic acid and fulvic acid were provided from International Humic Acid Association (Florida, USA). attapulgite (ATP), xanthan gum (XG) and aminosilicone oil (ASO) were provided by Fufeng Biotechnology Co., Ltd (Inner Mongolia, China). Waste milk was from Bright Dairy & Food co., ltd (Shanghai, China), with main components of 80.5(±0.6)% water, 6.8(±0.4)% fat, 4.8(±0.3)% protein, and 6.4(± 0.5)% lactose. Chickweed seeds were provided from Jiaoyan Seed Co., Ltd (Qingdao, China). Acidic soil was collected from a typical acidic soil district of Guigang in Guangxi Province (China). Soybean-composted organic fertilizer (COF, organic substances content of 40%, moisture of 30%, total NPK of 4%) was bought from LvDun Agriculture Co., Ltd (Shandong, China). Deionized water was used throughout the work except pot experiments.

### Waste milk humification and preparation of slow-release nano fulvic-like acid fertilizer
Humification tests were initiated by dosing PS (2–6 g) and KOH (2–6 g) to 50 mL waste milk at ambient temperature of 22 ± 1 °C with stirring. 0.5 mL of solution was taken at predetermined time intervals (10, 20, 30, 45, 60, 90, and 120 min) to analyze time-dependent evolution of fluorescent fractions in treated waste milk. Optimal treatment condition for humification was determined by evaluating effects of both dose and reaction time on the generation of HLA/FLA. The system temperature was observed to rise within the initial several minutes and monitored during the whole process. The liquid product (50 mL) under the optimal treatment condition (4 g PS and 4 g KOH for 1 h) was mixed with ATP (25 g) and xanthan gum (0.1 g) and the resulting system was granulated to spheres with a diameter of about 3 mm. Subsequently, the spheres were air-dried to around 30% moisture, and then soaked in ASO (100 mL) for 1 min to improve the mechanical stability of the spheres. After that, the resulting spheres were air-dried to obtain 55 ± 0.1 g slow-release nano fulvic-like acid fertilizer for use.

### Extraction and quantification of HLA and FLA in the product
The quantitative analysis was performed according to the modified method of BS ISO 19822-2018[69]. The product solution at pH of around

10.5 was firstly shaken for 1 h at 100 g, and then centrifuged (4000 g) for 15 min to separate the non-humic substances (Sediment 1) from soluble humic acid (SHA, a mixture of HLA and FLA) in the supernatant. Subsequently, the SHA supernatant was acidified to pH 1 ± 0.1 with 6 M hydrochloric acid, centrifuged (4000 g) for 15 min, and finally standing overnight to distinguish FLA (in the supernatant) from HLA (Sediment 2). The product solution, sediments 1 and 2 were dried to constant weights at 60 °C and weighted. The amounts of HLA and FLA were calculated using Eqs. (6)–(9), respectively.

$$m_{SHA} = m_{product} - m_1 - m_2 \qquad (6)$$

$$m_{FLA} = m_{SHA} - m_{HLA} \qquad (7)$$

$$HLA\% = \frac{m_{HLA}}{m_{product}} \times 100\% \qquad (8)$$

$$FLA\% = \frac{m_{FLA}}{m_{product}} \times 100\% \qquad (9)$$

Where $m_{SHA}$, $m_{product}$ and $m_{HLA}$ were the dry weights of SHA, product, and HLA, respectively. $m_1$ and $m_2$ represented the weight of Sediment 1 and $K_2SO_4$ stoichiometrically calculated on the premise of complete PS decomposition. $m_{FLA}$ referred to the calculated dry weight of FLA. To be noted, the ash contents (less than 0.3%) either in the product or HLA sediment, determined according to BS ISO 19822-2018[69], were minor to be subtracted in Eqs. (6)–(7).

The remained FLA in the supernatant was purified according to the following procedure: firstly adsorbed to an acidified hydrophobic resin Supelite DAX-8, then desorbed with 0.1 M NaOH, and finally protonated by flowing through a IR120 hydrogen form exchange resin column.

### Radical identification and quenching tests
Radicals of •OH and $SO_4^-$• generated during the humification process were measured, with 5,5-Dimethyl-1-pyrroline-N-oxide (DMPO) as the spin-trapping agent, using an electron paramagnetic resonance spectrometer (EPR spectrometer). To determine the contributions of each radical, quenching tests were conducted by adding 10 mol/L TBA or EtOH to wasted milk before dosing PS and KOH.

### Slow-release behavior
The prepared slow-release nano fulvic-like acid fertilizer (30 g) was put into 1 L water (adjusting to initial pHs of 3 and 7 using hydrochloric acid and NaOH). 5 mL solution was sampled at different intervals until equilibrium. The pH variation of the aqueous system was monitored during the process.

The contents of released SHA were measured using UV-vis spectrophotometry at the maximum absorbing wavelength of 267 nm. Considering that the solubility of SHA was pH-dependent, the product solution at pHs of 3, 4, 5, 6, 7, 8 and 9 was filtered, quantified (using weight method), and diluted to obtain the standard curve. Similarly, the contents of released FLA were determined according to its standard curve and the maximum absorbing wavelengths of 241 nm. To be noted, a standard curve of FLA was used in the whole pH range of 3–9 because its solubility was relatively not pH-dependent. The total loading amounts of FLA was calculated to be 1839 mg (Note S4) and available SHA amounts at pHs of 3, 4, 5, 6, 7, 8 and 9 in 30 g slow-release nano fulvic-like acid fertilizer were provided in Table S6.

### Pot experiment
To evaluate the remediation ability of the slow-release nano fulvic-like acid fertilizer on soil acidification, pot experiment was conducted using acidic soil (pH 5.0) with seven different treatments in triplicate:

waste milk, Control, ATP, $K_2SO_4$, COF, Product, and slow-release nano fulvic-like acid fertilizer. In each pot (height of 8.5 cm, bottom width and length of 7 cm, top width and length of 10 cm), acidic soil (120 g soil +280 g sand) was mixed with suitable amount of water to obtain a humidity of approximately 30%. Then 8 chickweed seeds were planted evenly at the soil depth of approximately 1 cm. The waste milk (1.4 mL), ATP (0.7 g), $K_2SO_4$ (0.14 g, equal to the content in 1.4 mL product), COF (1.5 g), product (1.4 mL), and slow-release nano fulvic-like acid fertilizer (1.5 g, made of 1.4 mL product and 0.68 g ATP) were spread evenly to the soil surface, respectively. During the whole growth period, 50 mL tap water was evenly sprayed onto the soil every three days. After harvest on day 21, the height, root length, fresh weight, dry weight, leaf area (leaf length × width), and germination rate of plants in each pot and chlorophyll contents of all leaves were recorded. Additionally, for each treatment, a total of 150 g soil was collected from triplicate pots (50 g each) for physicochemical properties (i.e. pH, total and available NPK, and soil organic matter) analyses. More detailed information about the pot tests were provided in Fig. S9.

### Characterizations
Contents of protein, fat, lactose, and total solids in waste milk were measured with an ultrasonic milk analyzer (Lactoscan Model MCC50, Milkotronic Ltd) by aspirating 15 cm³ milk sample per measurement via the analyzer according to the manufacturer's operational manual[70]. To preliminarily analyze the compositional changes of waste milk during the humification, three-dimensional Excitation-Emission matrix fluorescence (3D-EEM) spectrum was measured using a fluorescence spectrophotometer (F-7000, Hitachi High-Technologies Co., Japan). EEM results were quantitatively analyzed using FRI method (Note S2). Gel permeation chromatography (GPC, Waters-Wyatt, USA) was used to compare the molecular weight distribution of the filtered product solution and untreated waste milk. To compare its functional groups and element components with standard HLA/FLA, the freeze-dried product was characterized by Fourier transform infrared spectrometer (FTIR, NEXUS-670, Nicolet Co., USA), X-ray photoelectron spectroscopy (XPS, ESCALAB 250, Thermo-VG Scientific Co., USA) and solid-state $^{13}C$ nuclear magnetic resonance (NMR, Bruker AVANCE III, Karlsruhe, Germany). TOC was determined with a TOC analyzer (TOC-L, Shimadzu Co., Japan).

The morphologies of product and slow-release nano fulvic-like acid fertilizer were observed by a scanning electron microscopy (SEM; S4800, Hitachi High-Technologies Co., Japan, 100 to 300,000 times of magnification) with an accelerating voltage of 0.5-30 kV. X-ray diffraction (XRD; D/max-2550VB/PC, Rigaku Co., Japan) was applied to investigate the crystal structure of samples. PS concentration during humification was measured by a modified iodimetry spectrophotometric method at 352 nm[71,72].

Soil pH was measured in a 1:2.5 soil/water slurry. Determination of total N, P, and K (TN, TP, and TK) in soil samples were according to Kjeldahl digestion, molybdenum antimony colorimetry and the flame photometry, respectively[73]. Available N, P, and K (AN, AP and AK) in soil samples were measured using alkali diffusion, $NaHCO_3$ extraction colorimetry, and flame photometry, respectively[74]. Soil organic matter (SOM) was extracted by the $K_2CrO_7$-$H_2SO_4$ heat treatment, and residue was determined using $FeSO_4$ titration of potassium dichromate method[74].

### Statistical analysis and reproducibility
All experiments in the main text and supplementary information were performed in triplicate or more independently with similar results, with the quantitative results were presented as mean ± standard deviation. To demonstrate the imaging performance, representative imaging results were presented in the manuscript. Statistical analyses between groups were analyzed with Student's t-test ($p < 0.05$) using SPSS package version 22 (SPSS, Inc., Chicago,

IL). Significant differences for multiple comparisons for single point experiment were determined by one-way ANOVA with Tukey's HSD test as indicated in figure legends. All statistical tests used were two-sided in this study.

### Reporting summary

Further information on research design is available in the Nature Portfolio Reporting Summary linked to this article.

## Data availability

The authors declare that all data of this study are available within the article and the supplementary information. Source data are provided with this paper. Any additional data can be obtained from the corresponding author upon reasonable request. Source data are provided with this paper.

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

## Acknowledgements

This work was supported by the National Natural Science Foundation of China (Grant No. 52000023, Y.Z., 52370129, D.C., and 52000025, D.W.),

the Fundamental Research Funds for the Central Universities (Grant No. 22D111317, D.C.), the Key R&D Program of Guangxi Province (Grant No. 2022AB27060, Y.Z.), the Key R&D Program of Guangdong Province (Grant No. 2020B0202010005, D.C.), the Key R&D Program of Inner Mongolia Autonomous Region (Grant No. 2021GG0300, D.C.), the Key Program (Achievement Transformation) of "Revitalizing the City by Science and Technology" of Hulun Buir (Grant No. 2022HZZX008, H.X.), and the Key R&D Program of Shandong Province (Grant No. 2022SFGC0302, D.C.).

## Author contributions

Y.Z., D.C. and S.S. designed the experiments. Y.C. and B.F. conducted the experiments and prepared the manuscript. D.C. and H.X. supervised the study. C.W., P.Z., D.W. and N.Z. provided constructive suggestions for the manuscript revision.

## Competing interests

The authors declare no competing interests.
