## [Peer Review File · Nature Communications]

Reviewers' Comments:

Reviewer #1:

Remarks to the Author:

This manuscript provides a practical and effective way to recycle waste milk, which is supposed to be transformed into organic fertilizers participated by free radical. It is relatively rounded system including the humification process and its application in agriculture, which make it meaningful. After reading it carefully, I wish to point out some points as below for further consideration; these suggestions should have some helps on improving the paper.

1. Introduction. The necessity of recycling waste milk should be noted in the first paragraph.
2. Line 54. Please cited references in proper places.
3. The figure 1 should be arranged to make it more logical and readable.
4. The "4 g PS/4 g KOH and reaction time of 60 min" is regarded as the optimal humification condition. In this manuscript, the different dosage of PS and KOH were investigated, but the treatment of 4 g PS/4 g KOH was the only one that PS and KOH were mixed at the ratio of 1:1, what if the 6 g PS/ 6g KOH or others?
5. In this manuscript, the weight method is used to quantify HLA and FLA, why the TOC instrument was not selected?
6. Results and discussion. More sentences are related to results observed in this study, less discussions are presented, which make it plain.
7. The bacterial abundance of blank and SRNFF was exhibited in figure 6, what about other two treatments?
8. The graphical abstract should be improved beautifully.
9. Suggest the author supply the detailed significance of KOH addition in introduction.
10. Line 68, why more long-chain fatty acids generated after applying this artificial method? And what's the meaning of -COOH appearance? More reference should be replenished in the results and discussions.
11. Were there any limitations to the application of SRNFF?

Reviewer #2:

Remarks to the Author:

Manuscript review of NCOMMS-23-09093-T "Radical-induced rapid humification of waste milk and the performance of derived slow release fulvic-like fertilizer" by Zhu et al.

General Comments- This paper describes the production, characterization, use and potential toxicity of slow-release fertilizer material generated from the radical induced rapid humification of waste milk. The authors note that up to 100 million tons (20%) of produced raw milk is wasted each year and that disposal is both resource inefficient and environmentally damaging (although I'm not sure I get the last part of this; damaging how?). Given that the material is nutrient rich, there is interest in using it as a fertilizer but straight amendment to soil is highly problematic given the chemical composition. Therefore, the authors sought to develop an artificial humification process through an advanced oxidation process using base-activated persulfate to convert perishable organics to more stable humic acid like and fulvic acid like components. The produced material was characterized by a wide range of orthogonal techniques, including EPR spectrometry, UV-Vis, FTIR, GPC, and electron microscopy. The impact on chickweed growth, microbial community composition and earthworm survival was then evaluated. In short, the authors demonstrated successful production of a slow release fulvic-like fertilization that increased plant growth (relative to no fertilizer at all, not against conventional fertilizers), microbial diversity, and was non-toxic to earthworms. The paper topic is timely, adds some information to our understanding how to reuse a significant waste material in support of a circular agricultural economy, and is within the scope of Nature Communications. The synthesis and characterization techniques are particularly robust, although I have some concerns over the design of the plant and earthworm studies. As currently written, I have a number of general concerns and a range of specific comments that prevent me from recommending this paper for publication in its current form. With significant amount of effort, the authors may be able to address these concerns and if that is done, the paper could be acceptable for publication.

1. In the abstract, I'd delete the lines 28-30. This type of background information isn't needed in the abstract; just get right to what you did and what you found. Put more text in the results. Also, more detail is needed on the plant and worm studies, including dose, duration, and controls. The text on soil bacteria impacts is quite vague; be more specific and detailed.

2. The QA/QC on all of your analytical techniques is missing; measures of linearity, precision and accuracy would be typical. This can all be added to the SI. Also, the controls for your plant and earthworm study are unclear; add details. For the plant study, I'd argue that you need a conventional fertilizer control for comparison; your design is missing this. Please comment. Last, your methods on line 438 are very unclear; add some details there and in the SI.

3. Figure 6 and tables 2-3 need a statistical analysis.

4. I think some discussion of economics and possibly of the need for a life cycle analysis to truly understand the cost and benefits of this approach is needed. You don't need to actually do the LCA but I think you need to introduce the concept and say why it needs to be done.

5. The overall writing and presentation is good in some/most places but cumbersome and awkward in others. A thorough sentence-by-sentence copyediting by someone with the appropriate expertise is needed to sharpen the presentation.

Specific Comments-

Line 38- the comparison to a blank is not really relevant; you need comparison to a conventional fertilizer

Line 51- give more detail on how this causes environmental pollution

Lines 55-56, 67- awkward; rephrase

Line 77- delete "To be noted"

Line 83- replace "bricks" with "building blocks"

Figure 1- is there any way to statistically evaluate this data?

Line 140-give the standard deviations in parentheses

Line 238- "plenty"? Be more specific

Figure 6- define the treatments in the legend; add a statistical analysis to panels B-G

Table 2-3- add a statistical analysis

Line 374- how was this composition determined? Methods? Instruments? Error and QA/QC?

Line 382- "certain period"? Be more specific

Line 385- define all acronyms at first use (ATP)

Line 392- should be "centrifuged"

Line 430- what is the pH at the start; add soil characterization to the SI

Line 438- this is too vague; give some method details here

Point-by-point response to the referees' comments

We sincerely thank the referees for their thorough review and valuable comments, which have significantly improved the manuscript. We also extend our gratitude to the editor for the offered opportunity to revise the manuscript. In response to the referees' comments, we have revised the manuscript and highlighted the changes in red in the revised version. The point-by-point response to the referees' comments are outlined below. Please note that the line numbers referenced herein are those of the tracked version of the revised manuscript.

Reviewer #1 (Remarks to the Author):

This manuscript provides a practical and effective way to recycle waste milk, which is supposed to be transformed into organic fertilizers participated by free radical. It is relatively rounded system including the humification process and its application in agriculture, which make it meaningful. After reading it carefully, I wish to point out some points as below for further consideration; these suggestions should have some helps on improving the paper.

Response: We appreciate the reviewer's positive feedback on our work regarding the significance of proposed humification process and products' agricultural application. To address related concerns, a point-by-point response to those specific comments is provided below.

1. Introduction. The necessity of recycling waste milk should be noted in the first paragraph.

Response: Thanks for the suggestion. We have clarified the necessity of recycling waste milk in the first paragraph. The revised sentences in L38-42 were as follows:

“By far, the majority of WM was disposed directly to receiving water bodies, inducing water eutrophication and threatening aquatic life³. On the other hand, WM was released into sewage system and discharged after conventional wastewater treatment, which may cause greenhouse gas emission and resource loss⁴.”

2. Line 54. Please cited references in proper places.

Response: We assume this comment is asking for adding references on waste milk’s recycling potential in agriculture, and thus two relevant references have been added to the end of the sentence in L54 (L44 in revised version) as follows.

Added references:

6. Li, Y. et al. Win-win for monosodium glutamate industry and paddy agriculture: Replacing chemical nitrogen with liquid organic fertilizer from wastewater mitigates reactive nitrogen losses while sustaining yields. *Journal of Cleaner Production* **347**, 131287 (2022).

7. Drangert, J.-O. & Kjerstadius, H. Recycling–The future urban sink for wastewater and organic waste. *City and Environment Interactions* **19**, 100104 (2023).

3. The figure 1 should be arranged to make it more logical and readable.

Response: Thanks for pointing this out. Figure 1 has been rearranged by dividing into two parts, the upper reflected the dose effect of PS and KOH on humification of waste milk, while the lower represented the time-dependent 3D EEM evolution of treated WM at optimal dosage conditions. The revised figure 1 was as followed.

Fig. 1 3D-EEM of WM treated by KOH/PS under different conditions. Experimental conditions: WM volume=50 mL, PS or KOH dosage=0-6 g, ambient temperature=22±1 °C, and reaction time=0-2 h.

4. The “4 g PS/4 g KOH and reaction time of 60 min” is regarded as the optimal humification condition. In this manuscript, the different dosage of PS and KOH were investigated, but the treatment of 4 g PS/4 g KOH was the only one that PS and KOH were mixed at the ratio of 1:1, what if the 6 g PS/6 g KOH or others?

Response: This is a good point. We were also concerned about the effect of the dosage ratio on humification, and we had explored the dosage effect in a wider range than displayed in the manuscript. To address your concern, the EEM images at 2 g PS/2 g KOH, 6 g PS/6 g KOH, and 8 g PS/8 g KOH were provided in Fig. S2, wherein the one at 6 g PS/6 g KOH was also added to Fig. 1 for comparison. Overall, the results

indicated that 4 g PS/4 g KOH was the optimal humification condition compared with others at ratio of 1:1. The added Fig. S2 was as followed. The decrease trend at overdoses (above 4 g) of PS and KOH may be due to self-quenching of radicals or enhanced decomposition of the precursors/humification product. The explanation has been added to the manuscript in L127-128.

Fig. S2 3D-EEM results of treated WM at different dosages of PS and KOH with the molar ration of 1:1. Experimental conditions: ambient temperature= 22 ± 1 °C, reaction time=30 min.

5. In this manuscript, the weight method is used to quantify HLA and FLA, why the TOC instrument was not selected?

Response: This is a good question. To the best of our knowledge, TOC method has been used for determining HA and FA concentrations^{1,2}, mainly reflecting the relative transformation ratio of available organic carbon into HA/FA-carbon and the corresponding humification degree. Additionally, TOC method may be also used to represent the concentrations of certain extracted or commercial standard HA/FA³. Comparatively, the weight method was seemed to be reasonable to determine HLA/FLA yields in products, considering the C contents therein varied greatly according to raw materials and humification processes. In fact, the weight method has been conceived as a standard quantitative method according to international standard

ISO 19822⁴, and also widely adopted especially in prior artificial humification studies⁵⁻⁷. Therefore, the weight method was used in this work and the detailed data were provided in Table S2. Additionally, the TOC variation was also evaluated and provided in Table S3.

References:

- [1] Zc, Z., Tx, Y., Zm, W., Li, Y.J. & Yq, W. Effects of exogenous protein-like precursors on humification process during lignocellulose-like biomass composting amino acids as the key linker to promote humification process. *Bioresource technology* **291**, 121882 (2019).
- [2] Wu, J., et al. How does manganese dioxide affect humus formation during bio-composting of chicken manure and corn straw? *Bioresource Technology: Biomass, Bioenergy, Biowastes, Conversion Technologies, Biotransformations, Production Technologies* **269** (2018).
- [3] Yao, Y., et al. Molecular composition of size-fractionated fulvic acid-like substances extracted from spent cooking liquor and its relationship with biological activity. *Environ. Sci. Technol.* **53**, 14752–14760 (2019).
- [4] BS ISO 19822. Fertilizers and soil conditioners-determination of humic and hydrophobic fulvic acids concentrations in fertilizer materials. (2018).
- [5] Shao, Y., et al. Hydrothermal humification of lignocellulosic components: Who is doing what? *Chemical Engineering Journal* **457**, 141180 (2023)
- [6] Wang, R., Li, D., Zheng, G., Cao, Z. & Deng, F. Co-production of water-soluble humic acid fertilizer and crude cellulose from rice straw via urea assisted artificial humification under room temperature. *Chemical Engineering Journal* **455**, 140916 (2023)

[7] Yang, F., Zhang, S., Cheng, K. & Antonietti, M. A hydrothermal process to turn waste biomass into artificial fulvic and humic acids for soil remediation. *Science of the Total Environment* **686**, 1140-1151 (2019).

6. Results and discussion. More sentences are related to results observed in this study, less discussions are presented, which make it plain.

Response: Thanks for your valuable comment. As suggested, we have added some contents to the **Results** section. The added information are as follows:

L145-149: On the other hand, Table S3 showed that the organic carbon loss was 20.5% during the whole process, which was much lower compared with composting of food waste (30-60%)⁴⁷. That is, this emerging humification process showed potential advantages of promoting product's fertilization effect and mitigating carbon emission over composting.

L197-199: These active groups were reported to contribute to HLA/FLA's function in exerting hormone-like effect in plant or improving the retention capacity of soil water and nutrient⁵⁵.

L215-220: This was reasonable considering previously reported high reaction rates (10^7 - 10^9 $M^{-1} s^{-1}$) between radicals and amino acids^{58,59}. Therefore, both radicals may contribute to the proposed multiple reactions (eqs. 1-4) and play a key role in the humification of WM, which was consistent with the reported oxidation (e.g. hydroxylation and carboxylation)⁵⁹ and polymerization (e.g. Maillard reaction) effects of $\bullet OH$ and $SO_4^{\bullet -}$ ^{60,61}.

L230-233: As for the heat intensively released during 60 min humification, that would be easily gathered and recycled for PS activation in industrial application, thus potentially saving the dosage of KOH for activation.

L251-253: Generally, FLA with lower molecular weights were more easily assimilated

by plant⁶⁴, therefore the as-prepared product herein probably possessed a good promotion effect on plant growth.

L281-282: In other words, the product could be seemed as a kind of combined organic-inorganic fertilizer containing both K⁺ and HLA/FLA⁶⁷.

L318-320: Such adsorption was likely owing to the aforementioned hydrogen bonds and intercalation among ATP and FLA/HLA in Fig. 4E and 4F.

L328-332: Importantly, the root length of the chickweeds with SRNFF was 66.7% higher compared with the Blank, probably because of the widely reported stimulation effect of FLA/HLA on root elongation^{70,71}. Similar plant-stimulation effects have been previously observed in studies using paper mill effluent-extracted or Fenton-derived FLA^{72,73}.

L352-358: Microbial alpha-diversity results (Table 1) demonstrated that abundance indices (Chao 1 and Sobs) and diversity index (Shannon) for the four groups showed the same order of WM>SRNFF>ATP>Blank (p<0.05), implying that these additives could significantly increase bacterial richness and diversity. The different clustering features of principal coordinates analysis (PCoA) (Fig. 6G) clearly demonstrated the variations induced by different additives.

L368-378: These results indicated that the addition of SRNFF was favorable to soil bacterial community by stimulating the abundance and diversity and enriching beneficial bacteria. For the WM group, the soil's microbial community was greatly changed compared to that of Blank, with *Micrococcaceae* shifted to the predominant bacteria (9.3% relative abundance vs. 0.9% in Blank, p<0.05). Besides, the aforementioned microbial richness and diversity were also significantly increased. Considering the wide existence of *Micrococcaceae* in food (e.g. meat and milk cheese) fermentation^{82,83}, these bacterial shifts suggested the fermentation of WM in soil, which

was damaging to the germination of seeds. In comparison to Blank, ATP addition generally showed relatively limited impact on soil microbial community, likely because ATP as an inorganic matter was not readily to involve in the microbial metabolism.

L390-392: The underlying structure-function relationship of the as-prepared SRNFF needs further investigation.

7. The bacterial abundance of blank and SRNFF was exhibited in figure 6, what about other two treatments?

Response: Thanks for your concern. The soil bacterial results of the other two treatments (WM and ATP groups) have been added to Fig. 6 and discussed in L352-358 and L368-378.

L352-358: Microbial alpha-diversity results (Table 1) demonstrated that abundance indices (Chao 1 and Sobs) and diversity index (Shannon) for the four groups showed the same order of WM>SRNFF>ATP>Blank ($p<0.05$), implying that these additives could significantly increase bacterial richness and diversity. The different clustering features of principal coordinates analysis (PCoA) (Fig. 6G) clearly demonstrated the variations induced by different additives.

L368-378: These results indicated that the addition of SRNFF was favorable to soil bacterial community by stimulating the abundance and diversity and enriching beneficial bacteria. For the WM group, the soil's microbial community was greatly changed compared to that of Blank, with *Micrococcaceae* shifted to the predominant bacteria (9.3% relative abundance vs. 0.9% in Blank, $p<0.05$). Besides, the aforementioned microbial richness and diversity were also significantly increased. Considering the wide existence of *Micrococcaceae* in food (e.g. meat and milk cheese) fermentation^{82,83}, these bacterial shifts suggested the fermentation of WM in soil, which was damaging to the germination of seeds. In comparison to Blank, ATP addition

generally showed relatively limited impact on soil microbial community, likely because ATP as an inorganic matter was not readily to involve in the microbial metabolism.

8. The graphical abstract should be improved beautifully.

Response: We agree with the reviewer that the graphical abstract requires huge improvement. While to be noted, the graphical abstract has been removed in our revised submission as requested by the editor.

9. Suggest the author supply the detailed significance of KOH addition in introduction.

Response: This is a very good point. We have added the detailed significance of KOH addition in the last paragraph of **Introduction** section (L79-85) as follows:

“Therein, the addition of KOH at a sufficient dose could theoretically produce a large amount of heat spontaneously during its dissolution process, and thus PS could be co-activated by base and heat. On the other hand, the solubility of main organics (e.g. proteins and amino acids with pKa values commonly lower than 7) in WM would be promoted at alkaline conditions³¹, probably contributing to accelerated mass transfer and reaction kinetics between radicals and organics³². Besides, K⁺ in the product was also a macronutrient essential for crops.”

10. Line 168, why more long-chain fatty acids generated after applying this artificial method? And what's the meaning of -COOH appearance? More reference should be replenished in the results and discussions.

Response: Thanks for your comment. After a careful check of related references, “long-chain fatty acids” in the original sentence has been changed to “alkyl groups” considering the latter was more accurate (seen in L171).

The strengthened signal of -COOH in product revealed that carboxylation reaction may be related to the key mechanism of humification, the occurrence of which was also widely reported in •OH and SO₄^{-•} based oxidation. On the other hand, carboxyl group

is one typical active group possessed in FA standard, thus its appearance in product evidenced product's structural similarity with FA. To notify this, the sentence "Therefore, both radicals may play a key role in the humification of WM, which was consistent with the reported radical oxidation and polymerization effects of •OH and SO₄^{-•}" in L215-220 has been changed to "This was reasonable considering previously reported high reaction rates (10⁷-10⁹ M⁻¹ s⁻¹) between radicals and amino acids^{58,59}. Therefore, both radicals may contribute to the proposed multiple reactions (eqs. 1-4) and play a key role in the humification of WM, which was consistent with the reported oxidation (e.g. hydroxylation and carboxylation)⁵⁹ and polymerization (e.g. Maillard reaction) effects of •OH and SO₄^{-•}^{60,61}."

Last, more references (i.e. References 34, 37-40, 47, 51, 52, 55, 58-61, 64, 67, 70-73, 75, 82, and 83) have been added to the results and discussions according to your suggestion.

11. Were there any limitations to the application of SRNFF?

Response: Thanks for the question. According to SRNFF's slow-release performance in water, system pH was elevated along with the release of HLA/FLA. The pH variation of acid soil with SRNFF in the pot experiment exhibited a similar trend. Therefore, SRNFF could be probably used as a typical acid soil conditioner while showed limited application potential in saline-alkali soil amendment. The corresponding discussion on its application was added as follow (**Discussion** section, seen in L411-414).

"Besides, results of pot experiment proved that the as-prepared SRNFF possessed significant growth-promoting and acidic soil-amending effects, indicating SRNFF's high application potential in improving productivity of acidic soil (up to 3.95 billion hectares) worldwide⁸⁴."

Reviewer #2 (Remarks to the Author):

Manuscript review of NCOMMS-23-09093-T “Radical-induced rapid humification of waste milk and the performance of derived slow release fulvic-like fertilizer” by Zhu et al.

General Comments-This paper describes the production, characterization, use and potential toxicity of slow-release fertilizer material generated from the radical induced rapid humification of waste milk. The authors note that up to 100 million tons (20%) of produced raw milk is wasted each year and that disposal is both resource inefficient and environmentally damaging (although I’m not sure I get the last part of this; damaging how?). Given that the material is nutrient rich, there is interest in using it as a fertilizer but straight amendment to soil is highly problematic given the chemical composition. Therefore, the authors sought to develop an artificial humification process through an advanced oxidation process using base-activated persulfate to convert perishable organics to more stable humic acid like and fulvic acid like components. The produced material was characterized by a wide range of orthogonal techniques, including EPR spectrometry, UV-Vis, FTIR, GPC, and electron microscopy. The impact on chickweed growth, microbial community composition and earthworm survival was then evaluated. In short, the authors demonstrated successful production of a slow release fulvic-like fertilization that increased plant growth (relative to no fertilizer at all, not against conventional fertilizers), microbial diversity, and was non-toxic to earthworms. The paper topic is timely, adds some information to our understanding how to reuse a significant waste material in support of a circular agricultural economy, and is within the scope of Nature Communications. The synthesis and characterization techniques are particularly robust, although I have some concerns over the design of the plant and earthworm studies. As currently written, I have a number of general concerns and a range of specific comments that prevent me

from recommending this paper for publication in its current form. With significant amount of effort, the authors may be able to address these concerns and if that is done, the paper could be acceptable for publication.

Author response: We thank Reviewer 2 for the encouragement and advisory comments. A point-by-point response to the comments of Reviewer 2 is provided as follows.

(1) As for the concern the reviewer mentioned about the environmental damages of waste milk's direct dispose, related explanation and revise could be seen in specific comment 2.

(2) For the pot experiment design without the conventional fertilizer group, the explanation could be seen in specific comment 1.

1. In the abstract, I'd delete the lines 28-30. This type of background information isn't needed in the abstract; just get right to what you did and what you found. Put more text in the results. Also, more detail is needed on the plant and worm studies, including dose, duration, and controls. The text on soil bacteria impacts is quite vague; be more specific and detailed.

Response:

(1) In the abstract, I'd delete the lines 28-30. This type of background information isn't needed in the abstract; just get right to what you did and what you found.

Thanks for your kind suggestion. The sentences in lines 28-30 have been deleted. The abstract has been rewritten by adding more information on humification mechanism and the pot experiment according to your suggestion. The revised abstract (L17-29) was as followed.

“In this work, a novel artificial humification technology using KOH-activated persulfate (KOH/PS) was proposed for the rapid recycling of waste milk (WM) in agriculture. WM was converted into product rich in fulvic-like acid (FLA) and humic-

like acid in 1 h. Importantly, the humification was a spontaneous exothermic process and the system temperature could reach 61 °C in 10 min. Under the coactivation of PS by KOH and generated heat, $\bullet\text{OH}$ and $\text{SO}_4^{\bullet-}$ were produced and contributed to proposed oxidation and polymerization reactions. The product was mixed with attapulgite to fabricate a slow-release nano FLA fertilizer (SRNFF). Pot experiment results with chickweeds indicated that SRNFF exhibited yield promoting and root elongation effects. Moreover, SRNFF also showed significant acidic soil amendment effects via improving soil pH and beneficial bacteria (e.g. *Bacillus*, *Sinomonas* and *Ammoniphilus*) abundance. Overall, this study provided a rapid method for the recycling of waste food and highly-concentrated organic wastewater.”

(2) Put more text in the results.

As for the description of results, more information has been added in **Results** section as follows:

L145-149: On the other hand, Table S3 showed that the organic carbon loss was 20.5% during the whole process, which was much lower compared with composting of food waste (30-60%)⁴⁷. That is, this emerging humification process showed potential advantages of promoting product’s fertilization effect and mitigating carbon emission over composting.

L197-199: These active groups contributed to HLA/FLA’s function in exerting hormone-like effect in plant or improving the retention capacity of soil water and nutrient⁵⁵.

L215-220: This was reasonable considering previously reported high reaction rates (10^7 - $10^9 \text{ M}^{-1} \text{ s}^{-1}$) between radicals and amino acids^{58,59}. Therefore, both radicals may contribute to the proposed multiple reactions (eqs. 1-4) and play a key role in the humification of WM, which was consistent with the reported oxidation (e.g.

hydroxylation and carboxylation)⁵⁹ and polymerization (e.g. Maillard reaction) effects of $\bullet\text{OH}$ and $\text{SO}_4^{\bullet-}$ ^{60,61}..

L230-233: As for the heat intensively released during 60 min humification, that would be easily gathered and recycled for PS activation in industrial application, thus potentially saving the dosage of KOH for activation.

L251-253: Generally, FLA with lower molecular weights were more easily assimilated by plant⁶⁴, therefore the as-prepared product herein probably possessed a good promotion effect on plant growth.

L281-282: In other words, the product could be seemed as a kind of combined organic-inorganic fertilizer containing both K^+ and HLA/FLA⁶⁷.

L318-320: Such adsorption was likely owing to the aforementioned hydrogen bonds and intercalation among ATP and FLA/HLA in Fig. 4E and 4F.

L328-332: Importantly, the root length of the chickweeds with SRNFF was 66.7% higher compared with the Blank, probably because of the widely reported stimulation effect of FLA/HLA on root elongation^{70,71}. Similar plant-stimulation effects have been previously observed in studies using paper mill effluent-extracted or Fenton-derived FLA^{72,73}.

L368-378: These results indicated that the addition of SRNFF was favorable to soil bacterial community by stimulating the abundance and diversity and enriching beneficial bacteria. For the WM group, the soil's microbial community was greatly changed compared to that of Blank, with *Micrococcaceae* shifted to the predominant bacteria (9.3% relative abundance vs. 0.9% in Blank, $p < 0.05$). Besides, the aforementioned microbial richness and diversity were also significantly increased. Considering the wide existence of *Micrococcaceae* in food (e.g. meat and milk cheese) fermentation^{82,83}, these bacterial shifts suggested the fermentation of WM in soil, which

was damaging to the germination of seeds. In comparison to Blank, ATP addition generally showed relatively limited impact on soil microbial community, likely because ATP as an inorganic matter was not readily to involve in the microbial metabolism.

L390-392: The underlying structure-function relationship of the as-prepared SRNFF needs further investigation.

(3) Also, more detail is needed on the plant and worm studies, including dose, duration, and controls.

As for the pot and earthworm experiments, the detailed procedures of the pot and earthworm experiments were supplemented to **Methods** section “**Pot experiment**” (L500-523) with the corresponding experimental setups were added to Fig. S8 and Fig. S10 in SI.

(4) The text on soil bacteria impacts is quite vague; be more specific and detailed.

The discussion on soil bacteria impacts have been clarified and expanded (seen in L352-358 and L368-378) as follows.

L352-358: Microbial alpha-diversity results (Table 1) demonstrated that abundance indices (Chao 1 and Sobs) and diversity index (Shannon) for the four groups showed the same order of WM>SRNFF>ATP>Blank ($p<0.05$), implying that these additives could significantly increase bacterial richness and diversity. The different clustering features of principal coordinates analysis (PCoA) (Fig. 6G) clearly demonstrated the variations induced by different additives.

L368-378: These results indicated that the addition of SRNFF was favorable to soil bacterial community by stimulating the abundance and diversity and enriching beneficial bacteria. For the WM group, the soil's microbial community was greatly changed compared to that of Blank, with *Micrococcaceae* shifted to the predominant bacteria (9.3% relative abundance vs. 0.9% in Blank, $p<0.05$). Besides, the

aforementioned microbial richness and diversity were also significantly increased. Considering the wide existence of *Micrococcaceae* in food (e.g. meat and milk cheese) fermentation^{82,83}, these bacterial shifts suggested the fermentation of WM in soil, which was damaging to the germination of seeds. In comparison to Blank, ATP addition generally showed relatively limited impact on soil microbial community, likely because ATP as an inorganic matter was not readily to involve in the microbial metabolism.

2. The QA/QC on all of your analytical techniques is missing; measures of linearity, precision and accuracy would be typical. This can all be added to the SI. Also, the controls for your plant and earthworm study are unclear; add details. For the plant study, I'd argue that you need a conventional fertilizer control for comparison; your design is missing this. Please comment. Last, your methods on line 438 are very unclear; add some details there and in the SI.

Response:

(1) The QA/QC on all of your analytical techniques is missing; measures of linearity, precision and accuracy would be typical. This can all be added to the SI.

Thanks for your suggestion. The QA/QC in this work has been improved in three aspects: 1) The detailed information and diagram of pot and earthworm experiments have been provided as well as analysis procedure of soil properties. 2) The error bars of the results have been provided throughout the manuscript. 3) The statistical analyses were performed by one-way analysis of variance (ANOVA) and described in L564-570. Related statistical analysis has been supplemented in figures and tables throughout the manuscript.

(2) Also, the controls for your plant and earthworm study are unclear; add details.

As for the pot and earthworm experiments, the variation was discussed by comparing with the blank, which were specified both in L328-332 and L352-358. The

detailed procedures of the pot and earthworm experiments were supplemented to “**Pot experiment**” section (L500-523) with the corresponding experimental setups were added to Fig. S8 and Fig. S10 in SI.

L328-332: Importantly, the root length of the chickweeds with SRNFF was 66.7% higher compared with the Blank, probably because of the widely reported stimulation effect of FLA/HLA on root elongation^{70,71}. Similar plant-stimulation effects have been previously observed in studies using paper mill effluent-extracted or Fenton-derived FLA^{72,73}.

L352-358: Microbial alpha-diversity results (Table 1) demonstrated that abundance indices (Chao 1 and Sobs) and diversity index (Shannon) for the four groups showed the same order of WM>SRNFF>ATP>Blank ($p<0.05$), implying that these additives could significantly increase bacterial richness and diversity. The different clustering features of principal coordinates analysis (PCoA) (Fig. 6G) clearly demonstrated the variations induced by different additives. As for the microbial community composition, Fig. 6H showed that compared to the Blank,

(3) For the plant study, I’d argue that you need a conventional fertilizer control for comparison; your design is missing this.

As the effect of organic fertilizer is greatly dependent on the raw material, it should be necessary to use the WM-based conventional fertilizer as control. In fact, we have tried our best to seek the one made of WM but failed. Therefore, main discussion remained revolving in the comparison with the Blank group in the revised version. To briefly compare the fertilization of the product with conventional fertilizer, a preliminary pot experiment has been performed using a kind of soybean pulp-based conventional fertilizer as control. The result indicated that the product herein showed a higher yield (26.6%) for chickweeds than the soybean pulp-based seen in Fig. S9.

Overall, at this stage, it is not our intent to show that the product is superior than the conventional fertilizers, and our aim is simply to show the effectiveness of the product. After all, the focus of this research is chemical transformation of waste milk, and performance evaluation just plays a supporting role. Future research can be done to further compare our product with conventional ones.

(4) Last, your methods on line 438 are very unclear; add some details there and in the SI.

Methods of soil bacterial communities and soil physicochemical properties have been specified in L544-563.

Soil pH was measured in a 1:2.5 soil/water slurry. Determination of total N, P, and K (TN, TP, and TK) in soil samples were according to Kjeldahl digestion, molybdenum antimony colorimetry and the flame photometry, respectively⁹¹. Available N, P, and K (AN, AP and AK) in soil samples were measured using alkali diffusion, NaHCO₃ extraction colorimetry, and flame photometry, respectively⁹². Soil organic matter (SOM) was extracted by the K₂CrO₇-H₂SO₄ heat treatment, and residue was determined using FeSO₄ titration of potassium dichromate method⁹².

Soil microbial community was analyzed using 16S rDNA high-throughput sequencing technology. Specifically, DNA was extracted using the E.Z.N.A.® soil DNA Kit (Omega Bio-tek, Norcross, GA, U.S.) according to manufacturer's instructions. The hypervariable regions V3-V4 of the bacterial 16S rRNA gene were amplified with forward primer 338F (5'-ACTCCTACGGGAGGCAGCAG-3') and reverse primer 806R (5'-GGACTACHVGGGTWTCTAAT-3'). The polymerase chain reaction (PCR) mixture and amplification cycling conditions were described in our previous study⁹³. All samples were amplified in triplicate. The sequencing of purified PCR products was conducted through Illumina Miseq platform at Majorbio BioPharm

Biotechnology Co. Ltd. (Shanghai, China). The resulting sequences were firstly demultiplexed according to previous work⁹⁴ and de-noised following the DADA2 pipeline with recommended parameters. Raw sequence data were deposited to the NCBI Sequence Read Archive (accession number PRJNA946690).

3. Figure 6 and tables 2-3 need a statistical analysis.

Response: Thanks for pointing this out. As suggested, statistical analyses have been added to Figure 6 and tables. The related description on statistical analysis has been added in the last paragraph of **Methods** section (L564-570).

L564-570: Statistical analysis and reproducibility

All experiments were performed in triplicate, with the quantitative results were presented as mean \pm standard deviation. Data were processed with Microsoft Excel 2016, and figures were generated with Origin 2021. The statistical analysis was performed by one-way analysis of variance (ANOVA) using SPSS package version 22 (SPSS, Inc., Chicago, IL). The statistical data of high-throughput sequencing was analyzed on the Majorbio cloud platform at <https://www.majorbio.com>.

4. I think some discussion of economics and possibly some discussion of the need for a life cycle analysis to truly understand the cost and benefits of this approach is needed. You don't need to actually do the LCA but I think you need to introduce the concept and say why it needs to be done.

Response: Thanks for your suggestion. The cost of this technology was generally compared with conventional composting (Table S5). The cost difference mainly occurred in the treatment unit other than collection, transportation and utilization ones. This technology could save operation cost by 4 RMB/ton because the saving costs (74 RMB/ton) of labor, plant/equipment depreciation and maintenance could cover that (70

RMB/ton) of materials. The corresponding contents have been added to the manuscript (L419-424) and Table S5.

L419-424: The cost of this technology was generally compared with conventional composting (Table S5). The cost difference mainly occurred in the treatment unit other than collection, transportation and utilization ones. This technology could save operation cost by 4 RMB/ton because the saving costs (74 RMB/ton) of labor, plant/equipment depreciation and maintenance could cover that (70 RMB/ton) of materials.

Table S5 Operation cost comparison of this technology with conventional composting

Cost type (RMB/ton organic fertilizer)	This work (RMB)	Conventional composting (RMB)
Material	150	80
Labor	20	50
Electricity	10	10
Equipment depreciation	20	40
Maintenance	4	10
Plant depreciation	2	20
Package	50	50
Total	256	260

A brief comparison of carbon emission and energy consumption/reuse was performed to assess the environmental impacts of this technology. Added Table S6 and relevant discussion (L424-432) were as followed.

Table S6 Brief comparison of carbon emission and energy consumption between this technology with conventional composting

Items (per ton food waste)	This work	Conventional composting	Saving rate (%)
CO ₂ emission (kg)	36.6 ^a	94.7	61.35
Electricity consumption (kWh)	6.7	6.7	0
Heat released for reuse (kJ)	8.4	0 ^b	100

^a Assuming all the carbon loss in form of CO₂.

^b Too long duration to collect and reuse.

L424-432: Additionally, a brief comparison of carbon emission and energy consumption/reuse was performed to assess the environmental impacts of this technology. Table S6 illustrated that this technology could reduce CO₂ emission by 61.35% according to the TOC loss (20%) during WM's humification⁸³. Meanwhile, the intensively-generated heat (8.4 kJ/ton) was readily to collect for reuse. Overall, this novel humification technology showed advantages of cost-effective and environmentally friendly. In the future, further life cycle cost (LCC) and life cycle assessment (LCA) will be performed for the comprehensive evaluation of its application potential.

5. The overall writing and presentation is good in some/most places but cumbersome and awkward in others. A thorough sentence-by-sentence copyediting by someone with the appropriate expertise is needed to sharpen the presentation.

Response: Thanks for your kind suggestion. The editing has been polished throughout the manuscript and supporting information with the help of relevant experts.

Specific Comments-

1. Line 38- the comparison to a blank is not really relevant; you need comparison to a conventional fertilizer

Response: Thanks for your suggestion. We acknowledge the merit of this comment. As the effect of organic fertilizer is greatly dependent on the raw material, it should be necessary to use the WM-based conventional fertilizer as control. In fact, we have tried our best to search the one made of WM but failed. Therefore, main discussion remained revolving in the comparison with the Blank group in the revised version. To briefly compare the fertilization of the product with conventional fertilizer, a preliminary pot experiment has been performed using a kind of soybean meal-based conventional

fertilizer as the control. The result indicated that the product herein showed a higher yield (26.6%) for chickweeds than the soybean pulp-based seen in Fig. S9 as followed.

Fig. S9 Comparison of fertilization effects between SRNFF and conventional bean pulp-based organic fertilizer (COF). Experimental conditions: groups of Blank, COF, and SRNFF refer to 400 g soil alone, 400 g soil+1.5 g COF, and 400 g soil+1.5 g SRNFF, respectively. Each pot contained 8 chickweed seeds with harvest in 21 d. Each group was in triplicates and data were presented as mean \pm standard deviations.

Overall, at this stage, it is not our intent to show that the product is superior than the conventional fertilizers, and our aim is simply to show the effectiveness of the product. After all, the focus of this research is chemical transformation of waste milk, and performance evaluation just plays a supporting role. Future research can be done to further compare our product with conventional ones.

2. Line 51- give more detail on how this causes environmental pollution.

Response: Thanks for your suggestion. We have added more detailed description on how this causes environmental pollution to the revised manuscript (seen in L38-42) as follows:

“By far, the majority of WM was disposed directly to receiving water bodies, inducing water eutrophication and threatening aquatic life³. On the other hand, WM was released

into sewage system and discharged after conventional wastewater treatment, which may cause greenhouse gas emission and resource loss.”

3. Lines 55-56, 67- awkward; rephrase.

Response: Thanks for your suggestion. The sentences have been rephrased as follows:

L45-46: “However, those perishable organic components tended to ferment in soil and thus lower crops germination rate, limiting direct use of WM in farmland.”

L55-56: “The microbial-mediated humification usually needs several weeks with relatively low HLA/FLA yield.”

4. Line 77- delete “To be noted”.

Response: Thanks for your suggestion. " To be noted " in line 77 has been deleted.

5. Line 83- replace “bricks” with “building blocks”.

Response: We apologize for the mistake. As suggested, “bricks” has been replaced by “building blocks”.

6. Figure 1- is there any way to statistically evaluate this data?

Response: Thanks for your concern. In fact, the experiments corresponding to Figure 1 had conducted in triplicates, though only one among triplicate results were presented in Fig. 1. To address the concerns, the errors of the FRI results have been added to Table S1 to reflect the reproducibility of those EEM results. The p value has been added to L115 for comparison based on FRI results.

7. Line 140-give the standard deviations in parentheses.

Response: Thank you very much for pointing out this. The standard deviations have been added in parentheses. The revised version (L141-143) now read:

“Their yields were calculated to be 189(±1.8) mg HLA/g (18.9%), 453(±4.6) mg FLA/g (45.3%), and 642(±8.0) mg SHA/g (64.2%) respectively, according to eqs. (6-9) (Table

S2).”

8. Line 238- “plenty”? Be more specific.

Response: Thanks for pointing out this. It is true that SEM image was helpful in identifying surface structure instead of quantifying the substance on the surface, thus “plenty” here seems not appropriate. The sentence you mentioned has been changed to “In SRNFF, the products distributed in the pores and on the rods’ surface” (L260-261).

9. Figure 6- define the treatments in the legend; add a statistical analysis to panels B-G.

Response: Thank you very much for pointing out these omissions. The treatments have been specified in the revised legend of Fig. 6. Also, statistical analyses have been added to panels B-G. Besides, the related notification on statistical analysis has been added in L564-570. The revised Figure 6 was as follows.

Fig. 6 (A) Photographs, (B) plant height, (C) root height, (D) wet weight, and (E) dry weight of chickweed. (F) pH variation, (G) the principal co-ordinates analysis (PCoA) based on Euclidean distance algorithm, and (H) bacterial communities at genus level of soil in different groups. Experimental conditions: groups of Blank, ATP, WM, and

SRNFF refer to 400 g soil alone, 400 g soil+ 0.5 g ATP, 400 g soil+ 1 mL WM, and 400 g soil+1.5 g SRNFF, respectively. Each pot contained 8 chickweed seeds with harvest in 21 d. Each group was in triplicates and error bars represent the standard deviations. Different letters refer to significant differences at $p < 0.05$.

10. Table 2-3- add a statistical analysis.

Response: Thank you very much for pointing out this. A statistical analysis has been added as follows. Noted that the two tables had been wrongly numbered, which has been revised to tables 1 and 2.

Table 1 Richness and diversity estimators of the microbial communities in Blank and SRNFF groups.

Group	Chao 1	Sobs	Shannon	Simpson	Coverage
Blank	738±25d	738±25d	6.07±0.05b	0.0039±0.0002a	1±0.0a
WM	980±12a	980±13a	6.28±0.04a	0.0031±0.0003b	1±0.0a
ATP	803±27c	803.5±25c	6.12±0.05b	0.0034±0.0001a	1±0.001a
SRNFF	912±19b	913.9±20b	6.31±0.07a	0.0028±0.0004b	0.999±0.0a

Note: Data were expressed as the mean ± SD. Different letters in the same column refer to significant differences at $p < 0.05$.

Table 2 Mean length and weight of earthworms in soil with different treatments.

Group	Length (cm)		Δ length (cm)	Weight (g)		Δ weight (g)
	1 d	14 d		1 d	14 d	
Blank	11.6±0.1	12.7±0.2	1.1 ± 0.1c	10.4±0.2	11.4±0.2	1.0±0.02c
WM	10.7±0.2	11.6±0.2	0.9 ± 0.02d	11.1±0.2	12.1±0.2	1±0.04c
ATP	11.4±0.2	12.8±0.2	1.4 ± 0.05b	11.2±0.2	12.5±0.2	1.3±0.05b
SRNFF	10.8±0.3	12.5±0.2	1.7 ± 0.1a	11.2±0.2	13.7±0.2	1.5±0.02a

Note: Data were expressed as the mean ± SD. Different letters refer to significant differences at $p < 0.05$.

11. Line 374- how was this composition determined? Methods? Instruments? Error and QA/QC?

Response: Thanks for correcting our mistake here. The original data of WM components were provided by the instruction on the package, which ignored the

deviation caused by expiration. To correct the relevant data, we have analyzed the WM sample with an ultrasonic milk analyzer (Lactoscan Model MCC50, Milkotronic Ltd) in triplicates, achieving the main components as follows: 80.5(\pm 0.6)% water, 6.8(\pm 0.4)% lipid, 4.8(\pm 0.3)% protein, and 6.4(\pm 0.5)% lactose. The data have been renewed in the revised manuscript (L444-445). The optimal operation of the milk analyzer was determined through consultations with the manufacturer, with detailed methods and QA/QC provided in a prior study. Relevant introduction about the analyzer has been added in L512-514 (as follows).

L525-527: Contents of protein, fat, lactose, and total solids in WM were measured with an ultrasonic milk analyzer (Lactoscan Model MCC50, Milkotronic Ltd) according to a previously published procedure⁸⁶.

12. Line 382- “certain period”? Be more specific.

Response: Thank you very much for your concern. Specific reaction time and other detailed experimental design about the determination of optimal humification condition has been clarified (in L450-456) as follows:

“Humification tests were initiated by dosing PS (2-6 g) and KOH (2-6 g) to 50 mL WM at ambient temperature of 22 \pm 1 °C with stirring. 0.5 mL of solution was taken at predetermined time intervals (10, 20, 30, 45, 60, 90, and 120 min) to analyze time-dependent evolution of fluorescent fractions in treated WM. Optimal treatment condition for humification was determined by evaluating effects of both dose and reaction time on the generation of HLA/FLA. The system temperature was observed to rise within the initial several minutes and monitored during the whole process.”

13. Line 385- define all acronyms at first use (ATP).

Response: Thank you very much for your careful review. This kind of acronym errors have been checked carefully and revised throughout the manuscript.

14. Line 392- should be “centrifuged”.

Response: Sorry for the mistake. We have corrected the error.

15. Line 430- what is the pH at the start; add soil characterization to the SI.

Response: Thanks for your concern. The initial pH is 5.0, and the value has been added in line 502. Detailed method for soil characterization (i.e. microbial community and physicochemical properties analyses) has been added to the manuscript (L544-563) as follows:

“Soil pH was measured in a 1:2.5 soil/water slurry. Determination of total N, P, and K (TN, TP, and TK) in soil samples were according to Kjeldahl digestion, molybdenum antimony colorimetry and the flame photometry, respectively⁸⁹. Available N, P, and K (AN, AP and AK) in soil samples were measured using alkali diffusion, NaHCO₃ extraction colorimetry, and flame photometry, respectively⁹⁰. Soil organic matter (SOM) was extracted by the K₂CrO₇-H₂SO₄ heat treatment, and residue was determined using FeSO₄ titration of potassium dichromate method⁹⁰.

Soil microbial community was analyzed using 16S rDNA high-throughput sequencing technology. Specifically, DNA was extracted using the E.Z.N.A.® soil DNA Kit (Omega Bio-tek, Norcross, GA, U.S.) according to manufacturer’s instructions. The hypervariable regions V3-V4 of the bacterial 16S rRNA gene were amplified with forward primer 338F (5'-ACTCCTACGGGAGGCAGCAG-3') and reverse primer 806R (5'-GGACTACHVGGGTWTCTAAT-3'). The polymerase chain reaction (PCR) mixture and amplification cycling conditions were described in our previous study⁹¹. All samples were amplified in triplicate. The sequencing of purified PCR products was conducted through Illumina Miseq platform at Majorbio BioPharm Biotechnology Co. Ltd. (Shanghai, China). The resulting sequences were firstly demultiplexed according to previous work⁹² and de-noised following the DADA2

pipeline with recommended parameters. Raw sequence data were deposited to the NCBI Sequence Read Archive (accession number PRJNA946690).”

16. Line 438- this is too vague; give some method details here.

Response: Thank you very much for pointing out this omission. We add more necessary details about the methods of soil characterization to the manuscript (L544-563), the content of which has been provided below specific comment 15.

Reviewers' Comments:

Reviewer #1:

Remarks to the Author:

1.L2 : "slow release fulvic-like fertilizer" should be slow "release fulvic-like acid fertilizer"

2.Abstract: Please add key data such as FLA and HLA yields, as well as plant yields.

3.Please check the writing of $\text{SO}_4^{2-}/\text{SO}_4^{2-}/\text{SO}_4^{2-}$ and harmonize it throughout the text.

4.L24: "slow-release nano FLA fertilizer" Where is nano embodied?

5.L93: "This work highlights a rapid heat-free artificial humification technology for the recycling of waste food and highly concentrated wastewater in sustainable agriculture". This study only investigated the humification of WM by KOH/PS. Due to the extremely unique nature of WM, I think it is difficult to generalize it to waste food and highly concentrated wastewater.

6.L138: "Their yields were calculated to be $189(\pm 1.8)$ mg HLA/g (18.9%), $453(\pm 4.6)$ mg FLA/g (45.3%), and $642(\pm 8.0)$ mg SHA/g (64.2%) respectively, according to eqs. (6-9) (Table S2)". The author claims that this is the yield of FLA/HLA/SHA, which is completely misleading to readers. The yield should be the ratio of the product to the raw material. I carefully reviewed the author's calculation formula (6-9). To be precise, the value provided by the author should be the content of FLA/HLA in the solid product rather than the yield. This exaggerates the yield of FLA/HLA/SHA. In addition, the author assumes that PS is fully decomposed when calculating the value of m_2 , but it is uncertain whether PS is fully decomposed (Table S2). If PS is not fully decomposed, using 2.58g as the m_2 value will result in a higher calculated FLA/SHA content. More importantly, the author overlooked the quality of KOH added during the experimental process (4g). Even if OH⁻ is completely consumed during the experiment, the mass of the remaining K⁺ is as high as 2.79g. This also leads to a higher content of FLA/SHA calculated by the authors. In summary, I think the authors may have subjectively or unintentionally overstated the FLA/HLA/SHA yields.

7.L140: "Importantly, the yields in this work were much higher than the reported HLA (2.6-8%) and FLA (0.7-3%) results via composting of food wastes⁴¹⁻⁴³ or manure⁴⁴⁻⁴⁶." Because there is a significant difference in organic content and physicochemical properties between food waste or feces and WM, comparing it with food waste or fecal compost is completely meaningless. Even for comparison, it is necessary to compare with WM compost.

8.L142: "On the other hand, Table S3 showed that the organic carbon loss was 20.5% during the whole process, which was much lower compared with composting of food waste (30-60%)⁴⁷. That is, this emerging humification process showed potential advantages of promoting product's fertilization effect and mitigating carbon emission over composting." How is organic carbon loss measured and calculated? How carbon loss reflects the potential advantages of promoting product's fertilization effect.. In addition, carbon loss alone cannot reflect the mitigation of carbon emissions. Because the author added a large amount of chemical agents (KOH and PS) in the humification experiment of WM, the production process of these chemicals will also emit a large amount of carbon.

9.L148: "As shown in Fig. 2A, the composition of the freeze-dried product was investigated via Fourier transform infrared spectrometer (FTIR) analysis compared with fulvic acid (FA) standard." Why compare freeze-dried product instead of purified FLA with FA standard. In addition to FLA, freeze-dried product also contain components such as HLA. What is the significance of this comparison?

10.In this study, the author repeatedly mentioned the Maillard reaction and used it to explain many phenomena. However, whether or not the Maillard reaction occurred during the experiment, and if so, how strong the reaction was, were not monitored. The author should at least monitor the changes in the precursors and products of Maillard reaction during the reaction process to showed the occurrence of Maillard reaction and its contribution to humification.

11.L204: "Meanwhile, PS concentration decreased sharply to 58.5% within 2 min and then slowly to 15.6% at 30 min (Fig. S4)". Meanwhile, PS concentration decreased sharply to 58.5% of the initial concentration within 2 min and then slowly to 15.6% of the initial concentration at 30 min (Fig. S4)? According to the author's PS addition amount (4g/50mL WM), the theoretical initial PS concentration should be 80g/L, but the initial PS concentration measured by the author is less than 55g/L (Fig. S4). Why is the difference between the values measured by the author and the theoretical values so significant?

12.L206: "For quenching tests, tert-butanol (TBA) was used as a quencher for $\bullet\text{OH}$ ($k_{\bullet\text{OH}/\text{EtOH}} = (3.8-7.6) \times 10^8 \text{ M}^{-1} \text{ s}^{-1}$), while ethanol anhydrous (EtOH) for both $\bullet\text{OH}$ and $\text{SO}_4^{\bullet-}$ ($k_{\bullet\text{OH}/\text{EtOH}} = 1.6 \times 10^7 \text{ M}^{-1} \text{ s}^{-1}$, $k_{\text{SO}_4^{\bullet-}/\text{EtOH}} = 1.9 \times 10^9 \text{ M}^{-1} \text{ s}^{-1}$)⁵⁷." ($k_{\bullet\text{OH}/\text{EtOH}} = (3.8-7.6) \times 10^8 \text{ M}^{-1} \text{ s}^{-1}$) should be ($k_{\bullet\text{OH}/\text{TBA}} = (3.8-7.6) \times 10^8 \text{ M}^{-1} \text{ s}^{-1}$). TBA and EtOH, as two organic compounds, may directly interfere with the humification process, such as EtOH reacting with humification precursors (e.g., amino acids). How can the author eliminate this interference? In addition, if we assume that TBA and EtOH do not directly interfere with the humification process, it can be seen from Fig. 3C that there is no FLA/HLA generation in the treatment with TBA addition (quench only the $\bullet\text{OH}$ while retaining $\text{SO}_4^{\bullet-}$), which proves that $\text{SO}_4^{\bullet-}$ does not play a role in the humification process. This result is contrary to the conclusion drawn by the author.

13.L251: "For the application convenience and enhancement of HLA/FLA utilization efficiency, the humification product was mixed with ATP and XG to prepare SRNFF with diameter of approximately 3 mm." The author claims to have prepared SRNFF for the convenience of application and to improve the efficiency of HLA utilization. However, the author did not provide relevant evidence. I believe that the preparation of SRNFF is unnecessary for the following reasons: 1) Original liquid fertilizers can be applied to the soil with the irrigation water during irrigation, while solid SRNFF can only be fertilized through manual or mechanical sowing. So it seems that SRNFF is not convenient to apply. 2) Will SRNFF really improve the utilization efficiency of FLA/HLA? If so, please provide the data. 3) The granulation process and the chemicals (ATP, XG, and ASO) added during the granulation process will increase production costs. 4) The authors showed that "The loadings of SHA and FLA in SRNFF were determined to be 103.3 and 72.9 mg/g. (L291)". Please calculate the load rate of FLA, that is, how much FLA is loaded onto SRNFF in the liquid product and how much FLA still exists in the liquid. I roughly calculated based on the data provided by the author and found that the FLA loaded onto SRNFF was less than 50% of the original liquid product. Such a low load rate can cause waste of FLA. How to handle the remaining liquid after loading? 5) More importantly, the author should demonstrate the necessity of preparing SRNFF by comparing its effects on plant growth with original liquid products through pot experiments.

14.L320: "The fertilization effect of SRNFF". In this section, the author evaluated the fertility of SRNFF. However, the author did not compare the fertilizer efficiency of SRNFF with traditional fertilizers such as potassium fertilizer. This cannot prove the benefits of SRNFF. More importantly, SRNFF contains a large amount of K^+ , which is released into the soil in large quantities (the effective K content in SRNFF treatment is 2.45 times that of the blank treatment (Table S4)). Compared with the blank, the promotion of plant growth by SRNFF may be attributed more to the release of K^+ than to FLA/HLA. The author cannot prove that the target product FLA/HLA has a promoting effect on plant growth. The authors should at least prove that SRNFF has better fertilizer efficiency than traditional K fertilizer.

15.The author neglected the cost of the chemicals consumed in this technology when comparing it with other technologies (Table S5). The new technology described by the author consumes 80 kg of KOH and 80 kg of $\text{K}_2\text{S}_2\text{O}_8$ for each ton (1000 L) of WM processed. The cost of KOH and $\text{K}_2\text{S}_2\text{O}_8$ is much higher than the cost calculated by the author (256 RMB).

Reviewer #2:

Remarks to the Author:

Nice job responding to my comments

Point-by-point response to the referees' comments

We sincerely thank the referees for their thorough review and valuable comments, which have significantly improved the manuscript. We also extend our gratitude to the editor for the offered opportunity to revise the manuscript. In response to the editor and referees' comments, we have revised the manuscript and highlighted the changes in red in the revised version. The point-by-point response to these comments are outlined below. Please note that the line numbers referenced herein are those of the tracked version of the revised manuscript.

Response to Reviewer #1:

1. L2 “slow release fulvic-like fertilizer” should be “slow release fulvic-like acid fertilizer”

Response: Thanks for the suggestion. The “slow release fulvic-like fertilizer” has been replaced by “slow release fulvic-like acid fertilizer” in the title.

2. Abstract: Please add key data such as FLA and HLA yields, as well as plant yields.

Response: Thanks for the suggestion. These data have been added to the abstract as followed:

L19-20, P2: “At optimal dosages of 4 g KOH/4 g PS, WM was converted into product with 25.5% of fulvic-like acid (FLA) and 18.9% of humic-like acid (HLA) in 1 h.”

L25-31: “Pot experiment results with chickweeds indicated that applying product and SRNFF exhibited yield promoting (by 41.9 and 109%) and taproot elongation (by 24.9 and 58.2%) effects, implying their significant agricultural effects especially SRNFF. Therefore, the slow-release behavior could improve the efficacy of nutrients, and thus SRNFF was conceived as a value-added fertilizer. Moreover, soil pH was improved from 5 to 7.1 in 21 d with SRNFF, showing its significant acidic soil amendment effect.”

3. Please check the writing of SO₄[•]/SO₄⁻/SO₄[•] and harmonize it throughout the text.

Response: Thanks for your advice. The writings of sulfate radical you mentioned have been checked and consistently changed to “SO₄[•]” throughout the text.

4. L24: “slow-release nano FLA fertilizer” Where is nano embodied?

Response: Thanks for your concern. ATP was used as the carrier of humification product to fabricate the slow release fertilizer (SRNFF). Therein, ATP was conceived as a typical fibrous nanoclay with mean diameter of approximately 30 nm as seen in SEM image (Fig. 4C), which was consistent with reported work [1]. Fig. 5D showed that, after release, a lot of micro-nano pores appeared in SRNFF, indicating that the micro-nano structure of ATP could contribute to the slow release property (Fig. 5). Therefore, SRNFF was considered as a kind of nano fertilizer. The corresponding explanation about its nanostructure has been supplemented in Introduction section as followed:

L95-97: “Therein, ATP was a typical nanoclay consisting of nanorods with mean diameter of approximately 30 nm and widely used as a slow-release nanocarrier³³”.

References:

[1] Cai, D. *et al.* Controlling nitrogen migration through micro-nano networks. *Scientific Reports* **4**, 3665 (2014).

5. L93: “This work highlights a rapid heat-free artificial humification technology for the recycling of waste food and highly concentrated wastewater in sustainable agriculture”. This study only investigated the humification of WM by KOH/PS. Due to the extremely unique nature of WM, I think it is difficult to generalize it to waste food and highly concentrated wastewater.

Response: Thanks for your concern. Based on the main organic components (fat, protein, and lactose) of WM, the “waste food and highly concentrated wastewater” in

the manuscript has been changed to “dairy wastewater” (P2, L32 and P5, L101) according to your suggestion.

6. L138: “Their yields were calculated to be 189(\pm 1.8) mg HLA/g (18.9%), 453(\pm 4.6) mg FLA/g (45.3%), and 642(\pm 8.0) mg SHA/g (64.2%) respectively, according to eqs. (6-9) (Table S2).” The author claims that this is the yield of FLA/HLA/SHA, which is completely misleading to readers. The yield should be the ratio of the product to the raw material. I carefully reviewed the author's calculation formula (6-9). To be precise, the value provided by the author should be the content of FLA/HLA in the solid product rather than the yield. This exaggerates the yield of FLA/HLA/SHA. In addition, the author assumes that PS is fully decomposed when calculating the value of m_2 , but it is uncertain whether PS is fully decomposed (Table S2). If PS is not fully decomposed, using 2.58g as the m_2 value will result in a higher calculated FLA/SHA content. More importantly, the author overlooked the quality of KOH added during the experimental process (4g). Even if OH⁻ is completely consumed during the experiment, the mass of the remaining K⁺ is as high as 2.79g. This also leads to a higher content of FLA/SHA calculated by the authors. In summary, I think the authors may have subjectively or unintentionally overstated the FLA/HLA/SHA yields.

Response: (1) We appreciate your valuable comment. The “yield” has been replaced by “content” in the manuscript (L148, and L150 on P7).

(2) Thanks for pointing out the mistake, we rechecked the calculation and the FLA/SHA contents were unintentionally overstated mainly owing to the wrong calculation of K₂SO₄ mass (m_2). In terms of PS and OH⁻ remained, their residues were calculated to be 0.8 g PS/L (corresponding to PS decomposition rate of 99% in Fig. S4) and 0.005 g OH⁻/L (final pH of 10.5) in the product. Therefore, the remaining PS and OH⁻ were minor to be subtracted. Considering the nearly complete decomposition of persulfate

anion to sulfate anion according to eqs.1-3, value of m_2 was recalculated to be 5.16 g in 50 mL product. Accordingly, the FLA/HLA/SHA contents in product have been recalculated to be 25.5, 18.9, and 44.4% and updated throughout the manuscript (L20 on P2, L149 on P7, L390 on P21, and Fig. 5A and B) and SI (Table S2).

7. L140: “Importantly, the yields in this work were much higher than the reported HLA (2.6-8%) and FLA (0.7-3%) results via composting of food wastes 41-43 or manure 44-46.” Because there is a significant difference in organic content and physicochemical properties between food waste or feces and WM, comparing it with food waste or fecal compost is completely meaningless. Even for comparison, it is necessary to compare with WM compost.

Response: We totally understand your concern on the significance of the comparison among different starting materials. To address your concern, we have deleted the comparison with manure composting results to avoid any misleading, while the comparison with food waste composting was kept. This was because WM was a typical liquid food waste (LFW) mainly containing nutrients of protein, fat, and lactose, though with less total organic substance (around 20%) than solid FW (around 40%). Generally, LFW was difficult to be converted into fertilizer through composting and related work had been rarely reported to the best of our knowledge. This technology was supposed to overcome the limitation and realize the rapid humification of liquid food waste, thus could be considered as an emerging alternative route of organic fertilizer production to composting. In a word, WM was a kind of FW with organic fertilizer as its final product in this work, thus we believe it is reasonable to compare with composting of FW. The revised sentence was as followed.

L150-152: “the contents in the product of this work were much higher than the reported HLA (2.6-8%) and FLA (0.7-4%) results via composting of food wastes⁴²⁻⁴⁴.”

8. L142: “On the other hand, Table S3 showed that the organic carbon loss was 20.5% during the whole process, which was much lower compared with composting of food waste (30-60%)⁴⁷. That is, this emerging humification process showed potential advantages of promoting product’s fertilization effect and mitigating carbon emission over composting.” How is organic carbon loss measured and calculated? How carbon loss reflects the potential advantages of promoting product’s fertilization effect. In addition, carbon loss alone cannot reflect the mitigation of carbon emissions. Because the author added a large amount of chemical agents (KOH and PS) in the humification experiment of WM, the production process of these chemicals will also emit a large amount of carbon.

Response:

(1) We appreciate your comments. TOC values of the WM and product were determined using TOC instrument after frozen-drying. The carbon loss was obtained via their TOC difference. The corresponding content has been added to the Methods section of the manuscript as followed.

L530, P27: “TOC was determined with a TOC analyzer (TOC-L, Shimadzu Co., Japan).”

(2) As for the relationship between carbon loss and fertilization effect of the product: compared with traditional composting, this method showed a lower carbon loss, which indicated that more organic carbon in starting materials could be potentially conserved in the product and sequestered in soil after application, promoting the soil fertilization. Accordingly, in order to clarify this issue, the sentence “That is, this emerging humification process showed potential advantages of promoting product’s fertilization effect and mitigating carbon emission over composting” has been changed to “That is, compared with composting, this technology could potentially conserve more organic carbon in the product and soil after application, promoting the soil fertilization” (L154-

156, P8).

(3) Thanks for pointing out this. All the “CO₂ emission” and “carbon emission” throughout the manuscript (L153 on P8, L412 and 414 on P22) and in Table S6 have been changed to “carbon loss”.

In terms of the carbon emission regarding the addition of chemical agents (KOH and PS): during the humification, KOH and PS were transformed to K₂SO₄ that could be used as fertilizer and potentially reduce the extra usage of chemical K fertilizer. The carbon emission during production of KOH and PS may be neutralized by that during production of saved K fertilizer. Therefore, their addition may be not a major concern in increasing carbon emission.

9. L148: “As shown in Fig. 2A, the composition of the freeze-dried product was investigated via Fourier transform infrared spectrometer (FTIR) analysis compared with fulvic acid (FA) standard.” Why compare freeze-dried product instead of purified FLA with FA standard. In addition to FLA, freeze-dried product also contain components such as HLA. What is the significance of this comparison?

Response: Thanks for your concern and suggestion. In this work, since the product was planned to be directly used as fertilizer other than extracted FLA, FTIR analysis of the product was conducted to screen active groups possibly contributing to its fertilization effect. Moreover, FTIR spectrum of the product was compared with WM to obtain brief structural variations, preliminarily exploring the humification mechanism. In addition, the FTIR spectrum of product was compared with FA standard in order to prove that the main active groups of FA also existed in product. Therefore, the comparison between product and FA was believed to be necessary.

According to your advice, the FTIR spectrum of FLA after extraction and freeze-drying has been added to Fig. 2a, which showed similar active groups with FA standard.

The related analysis has been added to the manuscript (P8, L173-175).

10. In this study, the author repeatedly mentioned the Maillard reaction and used it to explain many phenomena. However, whether or not the Maillard reaction occurred during the experiment, and if so, how strong the reaction was, were not monitored. The author should at least monitor the changes in the precursors and products of Maillard reaction during the reaction process to show the occurrence of Maillard reaction and its contribution to humification.

Response: Thanks for your concern and suggestion. Maillard reaction was known as a heat-induced “nonenzymatic browning reaction” widely reported in humification process. In our original manuscript, the occurrence of Maillard reaction had been speculated mainly according to the darkening color of the system and the enhancement of C-N (1410 cm^{-1}) and N-H (1563 cm^{-1}) peaks after humification in FTIR results (Fig. 2a).

As requested, to further verify the occurrence and strength of Maillard reaction, FTIR, EEM and UV spectra variation of samples have been analyzed during humification at different intervals (0, 2, 5, 15, 30, and 60 min). Firstly, we monitored the variation of FTIR spectra and supplemented to Fig. 2a. Notably, the characteristic peaks of C-N (1400 cm^{-1}), N-H (620 and 1550 cm^{-1}) and aromatic -OH (1100 cm^{-1}) became stronger within initial 30 min and then (from 30 to 60 min) stable. While the peaks of alcoholic -OH ($3200\text{-}3500\text{ cm}^{-1}$) and aldehyde C=O (1750 cm^{-1}) became weakened similarly. These results likely indicated a significant process of amine aldehyde condensation, dehydration and cyclization in Maillard reaction.

Secondly, the EEM fluorescence spectroscopy had been widely applied for characterizing the formation of Maillard products at Ex/Em of 340-370 nm/420-440 nm^[1], which was consistent with the characteristic EEM spectra in our product (Fig.

1n). According to Fig. 1(i-p), the characteristic fluorescence intensity was gradually strengthened in the first 60 min, which also evidenced the occurrence of Maillard reaction.

Besides, the UV absorbance at 420 nm (A_{420}) had been used as an indicator of the Maillard reaction to show the browning degree ^[2]. As shown in Fig. S2, A_{420} exhibited a similar trend to the EEM spectra, also suggesting the occurrence of Maillard reaction in the first 60 min.

Overall, the above analyses mainly provided indirect evidence of Maillard reactions due to the complexity of our reaction system. Considering this study mainly focused on evaluating the efficacy of this new technology in humification and the agricultural effectiveness of the product, contribution of Maillard reaction to humification will be deeply investigated in future. The related content has been added to the revised manuscript in P8-9, L158-178, P11, L217-218, and Fig. 2a.

References:

[1] Wang, Q. et al. Mechanistic insights into the effects of biopolymer conversion on macroscopic physical properties of waste activated sludge during hydrothermal treatment: Importance of the Maillard reaction. *Science of the Total Environment* **769**, 144798 (2021).

[2] Xia, X. et al. Formation of fluorescent Maillard reaction intermediates of peptide and glucose during thermal reaction and its mechanism. *J. Agric. Food Chem.* **71**, 8569-8579 (2023).

11. L204: “Meanwhile, PS concentration decreased sharply to 58.5% within 2 min and then slowly to 15.6% at 30 min (Fig. S4).”. Meanwhile, PS concentration decreased sharply to 58.5% of the initial concentration within 2 min and then slowly to 15.6% of the initial concentration at 30 min (Fig. S4)? According to the author's PS addition

amount (4g/50mL WM), the theoretical initial PS concentration should be 80g/L, but the initial PS concentration measured by the author is less than 55g/L (Fig. S4). Why is the difference between the values measured by the author and the theoretical values so significant?

Response: Thanks for your concern. The y-axis was actually the concentration of persulfate anion, and we were sorry for the misleading. To address this issue, we have clarified in the legend of Fig. S4 as followed.

“Fig. S4 Time-dependent variation of $S_2O_8^{2-}$ concentration during the process. Experimental conditions: 4 g PS and 4 g KOH in 50 mL WM, ambient temperature= 22 ± 1 °C. Error bars represent the standard deviations from triplicate tests.”

12. L206: “For quenching tests, tert-butanol (TBA) was used as a quencher for $\bullet OH$ ($k_{\bullet OH/EtOH} = (3.8-7.6) \times 10^8 M^{-1} s^{-1}$), while ethanol anhydrous (EtOH) for both $\bullet OH$ and $SO_4^{\bullet -}$ ($k_{\bullet OH/EtOH} = 1.6 \times 10^7 M^{-1} s^{-1}$, $k_{SO_4^{\bullet -}/EtOH} = 1.9 \times 10^9 M^{-1} s^{-1}$).” ($k_{\bullet OH/EtOH} = (3.8-7.6) \times 10^8 M^{-1} s^{-1}$) should be ($k_{\bullet OH/TBA} = (3.8-7.6) \times 10^8 M^{-1} s^{-1}$). TBA and EtOH, as two organic compounds, may directly interfere with the humification process, such as EtOH reacting with humification precursors (e.g., amino acids). How can the author eliminate this interference? In addition, if we assume that TBA and EtOH do not directly interfere with the humification process, it can be seen from Fig. 3C that there is no FLA/HLA generation in the treatment with TBA addition (quench only the $\bullet OH$ while retaining $SO_4^{\bullet -}$), which proves that $SO_4^{\bullet -}$ does not play a role in the humification process. This result is contrary to the conclusion drawn by the author.

Response: Thanks for your concern. We agreed that possible reactions between quenchers and humification precursors were significant concerns, which may be addressed in the following two perspectives: (1) theoretically, esterification reaction

was the most possible reaction between alcohol (TBA and EtOH) and certain components (protein, fatty acids, and amino acids) of WM. While the esterification reaction was reversible and commonly occurred with the catalysis of concentrated sulfuric acid or high temperature (>50 °C). Notably, the temperature only climbed to lower than 30 °C in quenching systems herein (Fig. 3D). Therefore, the reaction was difficult to occur owing to the lack of those conditions in this system.

(2) To be noted, the obvious shifts of fluorescent components in regions II and IV during humification (Fig. 1m) indicated that those components were key precursors involved in the humification, while the addition of TBA or EtOH to WM (without PS/KOH) rarely changed the fluorescence in regions II and IV (Fig. S5). The result indicated that TBA or EtOH did not interfere with those key precursors greatly. **The related discussion and Fig. S5 have been added to the manuscript (P11-12, L225-230) and SI.**

These evidences may address the concern to some extent. That is, TBA and EtOH were seemed to not interfere with the humification process by directly reacting with precursors.

As for the contributions of radicals, we agree your suggestion about the conclusion that •OH probably played a key role in the humification of WM. The contribution of •OH to the humification could be verified by EEM spectra with addition of TBA in Fig. 3c. Considering that a major part of $\text{SO}_4^{\cdot-}$ could be transformed into •OH especially under highly alkaline conditions according to eqs. (1-4), the EEM spectra comparison between addition of EtOH and TBA actually could prove the insignificant contribution of the remaining $\text{SO}_4^{\cdot-}$. In a word, •OH played the dominant role in humification of WM, while $\text{SO}_4^{\cdot-}$ showed insignificant direct contribution. Accordingly, the conclusion of “Therefore, both radicals may contribute to the proposed multiple reactions (eqs. 1-4) and play a key role in the humification of WM, which was consistent with the

reported oxidation (e.g. hydroxylation and carboxylation)⁵⁹ and polymerization (e.g. Maillard reaction) effects of •OH and SO₄^{-•60,61} has been changed to “Therefore, •OH generated through the proposed multiple reactions (eqs. 1-4) may play a key role in the humification of WM, which was consistent with the reported oxidation (e.g. hydroxylation and carboxylation)⁵⁸ and polymerization (e.g. Maillard reaction) effects of •OH^{59,60}”.

Finally, thanks for pointing out the mistake about the writing “ $k_{\text{OH/EtOH}} = (3.8-7.6) \times 10^8 \text{ M}^{-1} \text{ s}^{-1}$ ”, and it has been revised to “ $k_{\text{OH/TBA}} = (3.8-7.6) \times 10^8 \text{ M}^{-1} \text{ s}^{-1}$ ” (L223, P11).

13. L251: “For the application convenience and enhancement of HLA/FLA utilization efficiency, the humification product was mixed with ATP and XG to prepare SRNFF with diameter of approximately 3 mm.” The author claims to have prepared SRNFF for the convenience of application and to improve the efficiency of HLA utilization. However, the author did not provide relevant evidence. I believe that the preparation of SRNFF is unnecessary for the following reasons: 1) Original liquid fertilizers can be applied to the soil with the irrigation water during irrigation, while solid SRNFF can only be fertilized through manual or mechanical sowing. So it seems that SRNFF is not convenient to apply. 2) Will SRNFF really improve the utilization efficiency of FLA/HLA? If so, please provide the data. 3) The granulation process and the chemicals (ATP, XG, and ASO) added during the granulation process will increase production costs. 4) The authors showed that “The loadings of SHA and FLA in SRNFF were determined to be 103.3 and 72.9 mg/g. (L291)”. Please calculate the load rate of FLA, that is, how much FLA is loaded onto SRNFF in the liquid product and how much FLA still exists in the liquid. I roughly calculated based on the data provided by the author and found that the FLA loaded onto SRNFF was less than 50% of the original liquid

product. Such a low load rate can cause waste of FLA. How to handle the remaining liquid after loading? 5) More importantly, the author should demonstrate the necessity of preparing SRNFF by comparing its effects on plant growth with original liquid products through pot experiments.

Response:

(1) Thanks for your concern. The product was transformed to solid SRNFF mainly for three reasons: 1) convenience and low-cost of package, transportation, and storage because of the low moisture (10-30%) than liquid fertilizer (75-90%), 2) application independence on irrigation facilities which were generally lack in rainfall-rich districts (e.g. acid soil district in southern China), and 3) sustainable nutrient supply for higher fertilization effect. The related explanation has been added to **P3, L92-93**.

(2) and (5): Thanks for your valuable comments. The fertilization effect of product has been supplemented in pot experiment for comparison with SRNFF (**Fig. 6 and Fig. S10**), and yields (fresh weight) in SRNFF group was observed 47.1% higher than that in product group. This result indicated that SRNFF possessed a higher (47.1%) agricultural utilization efficiency compared with product. **The related discussion has been added to P18, L348-350 and L352-354.**

(3) During the granulation process of per ton SRNFF (30.3% moisture, prepared with 900 L product), the costs of added chemicals (500 kg ATP, 2kg XG, and 0.2 kg ASO) and granulation process were 232 (200, 30 and 2) and 50 RMB respectively, increasing total cost by 282 RMB. However, during package, transportation and storage, the cost of per ton SRNFF could save approximately 700 RMB (400 package +250 transportation +50 storage). Additionally, SRNFF of per ton (dosing for 2 mu field) could increase the incoming of crops (e.g. chickweed) by approximately 4000 RMB. The total value (4700 RMB) of increased incoming (4000 RMB) and saved cost (700

RMB) was much higher than the increased cost during granulation process (282 RMB). Thus, the granulation of product herein to produce SRNFF was seemed to be meaningful.

(4) Thanks for your concern. During the fabrication of SRNFF, all the product (50 mL) was used with no liquid left, and the granulated SRNFF were air-dried to 55 ± 0.1 g (with 30.3% moisture) before use. Therefore, no liquid product was wasted during granulation, corresponding to a 100% loading rate. The release ratio of FLA and SHA were recalculated and updated in Fig. 5a and b. We were sorry for not introducing the granulation steps clearly and detailed information for calculation in the original manuscript. The related description and calculation have been added in L450-453, P24 and Text S4 as followed.

L445-448, P24: Subsequently, the spheres were air-dried to around 30% moisture, and then soaked in ASO (100 mL) for 1 min to improve the mechanical stability of the spheres. After that, the resulting spheres were air-dried to obtain 55 ± 0.1 g SRNFF for use.

Text S4 Moisture, FLA and SHA loading calculations of SRNFF

① Dry weight of SRNFF (m_{SRNFF}^{dry}) using 50 mL product, 25 g ATP and 0.1 g XG:

$$m_{SRNFF}^{dry} = m_{product}^{dry} + m_{ATP} + m_{XG} = 13.24 \text{ g} + 25 \text{ g} + 0.1 \text{ g} = 38.34 \text{ g}$$

② Moisture of SRNFF:

$$\text{Moisture\%} = 1 - m_{SRNFF}^{dry} / m_{SRNFF} = 1 - 38.34 \text{ g} / 55 \text{ g} = 30.3\%$$

③ Content of FLA in SRNFF:

$$\text{FLA\%} = m_{FLA}^{dry} / m_{SRNFF} = 3.37 \text{ g} / 55 \text{ g} = 6.13\%$$

④ Total amount of FLA for release in 30 g SRNFF:

$$M_{FLA} = 30 \text{ g} \times 6.13\% = 183.9 \text{ g}$$

⑤ Content of SHA in SRNFF:

$$\text{SHA}\% = m_{\text{SHA}}^{\text{dry}}/m_{\text{SRNFF}} = 5.87 \text{ g}/55 \text{ g} = 10.67\%$$

⑥ Total amount of SHA in 30 g SRNFF available for release at pH 10.5:

$$M_{\text{SHA}} = 30 \text{ g} \times 10.67\% = 3.2 \text{ g}$$

Besides, considering the total amount of SHA available for release varied with system pH, accurate amounts of SHA at varied pHs of 3-9 were obtained via eq. (6) by determining corresponding m_1 gravimetrically. The detailed data could be seen in Table S7.

Table S7 m_1 and SHA amounts at varied pHs.

pH	m_1 (g)	SHA in 50 mL product (g)	SHA in 30 g SRNFF (g)
3	3.8	4.3	2.3
4	3.7	4.4	2.4
5	3.6	4.4	2.4
6	3.4	4.7	2.6
7	2.8	5.2	2.9
8	2.6	5.5	3
9	2.5	5.6	3.1
10	2.2	5.9	3.2

14. L320: “The fertilization effect of SRNFF”. In this section, the author evaluated the fertility of SRNFF. However, the author did not compare the fertilizer efficiency of SRNFF with traditional fertilizers such as potassium fertilizer. This cannot prove the benefits of SRNFF. More importantly, SRNFF contains a large amount of K^+ , which is released into the soil in large quantities (the effective K content in SRNFF treatment is 2.45 times that of the blank treatment (Table S4)). Compared with the blank, the promotion of plant growth by SRNFF may be attributed more to the release of K^+ than

to FLA/HLA. The author cannot prove that the target product FLA/HLA has a promoting effect on plant growth. The authors should at least prove that SRNFF has better fertilizer efficiency than traditional K fertilizer.

Response: Thanks for your valuable comments. We agree with your concern that a high content of K in SRNFF would potentially contribute to fertilization effect. Thus, a group of pot experiments with 0.14 g conventional K_2SO_4 fertilizer (equal to the K_2SO_4 amount in 1.4 mL product or 1.5 g SRNFF) was supplemented to evaluate its effect. The result indicated that product, SRNFF and conventional K_2SO_4 fertilizer could increase chickweed yields (fresh weight) by 109, 41.9 and 9.4% respectively compared to Blank. Therefore, SRNFF showed a better fertilization effect than conventional K_2SO_4 fertilizer, proving the key contribution of FLA/HLA. The corresponding content has been added to the manuscript in Lines 350-352 on page 18.

15. The author neglected the cost of the chemicals consumed in this technology when comparing it with other technologies (Table S5). The new technology described by the author consumes 80 kg of KOH and 80 kg of $K_2S_2O_8$ for each ton (1000 L) of WM processed. The cost of KOH and $K_2S_2O_8$ is much higher than the cost calculated by the author (256 RMB).

Response: Thanks for your concern. Considering that scarce example could be found in WM composting, the cost comparison was conducted between this technology and composting using pig manures as raw material. The cost of the technology (256 RMB) was calculated based on the practical industry with KOH and PS dosages of both 15 kg in producing per ton solid organic fertilizer (meeting NY525-2021 organic fertilizer standard of China) other than 80 kg in WM's humification. Notably, the lower dosages compared with WM herein was probably because the temperature was observed to rise more easily in solid manure owing to the bulk effect, lower moisture, and different

components. In solid manure, the activation of PS by heat (easily retained in solid bulk system) besides KOH was also proposed to promote the humification at low chemicals dosages. The chemical cost calculation has been notified in L408, P22 and Table S5 as followed.

Response to reviewer #2:

General comments: Nice job responding to my comments

Response: We appreciate your valuable comments and recognition on our work. The pot experiment with traditional organic fertilizer you suggested has been repeated in the updated version.

Reviewers' Comments:

Reviewer #1:

Remarks to the Author:

Report for NCOMMS-23-09093B

This manuscript provides an effective way to recycle waste milk, which is supposed to be transformed it into organic fertilizers participated by free radical. It is relatively rounded system including the humification process and its application in agriculture, which make it meaningful. But under normal circumstances, milk does not spoil, which calls into question the practical significance of the findings. If this finding is credible and feasible, should we spoil milk to make fertilizer? Is it worth it? So, we think this research method and results are credible, but the prospects for application are bleak. As the Nature Communications is the multidisciplinary journal that publishes high-quality research achievement, in this case, we suggest to reject this manuscript.

Reviewer #2:

Remarks to the Author:

Very nice job responding to the reviewer comments

Reviewer #3:

None

Point-by-point response to the referees' comments

We sincerely thank the referees for their thorough review and valuable comments, which have significantly improved the manuscript. We also extend our gratitude to the editor for the offered opportunity to accept the manuscript. The point-by-point response to the referees' comments are outlined below.

Reviewer #1 (Remarks to the Author):

This manuscript provides an effective way to recycle waste milk, which is supposed to be transformed it into organic fertilizers participated by free radical. It is relatively rounded system including the humification process and its application in agriculture, which make it meaningful. But under normal circumstances, milk does not spoil, which calls into question the practical significance of the findings. If this finding is credible and feasible, should we spoil milk to make fertilizer? Is it worth it? So, we think this research method and results are credible, but the prospects for application are bleak. As the Nature Communications is the multidisciplinary journal that publishes high-quality research achievement, in this case, we suggest to reject this manuscript.

Response: Thanks for your positive comments on the humification process and recycling significance in agriculture proposed in our work. As for the practical application prospects of this process in milk management, we believe that it is of significance since the waste milk has been up to 13% of the produced milk globally in 2009 (mentioned at the starting of our introduction section). Besides, this study may also provide an example for the rapid humification of dairy wastewater. Briefly, this work highlights a rapid heat-free artificial humification technology for the recycling of waste milk in sustainable agriculture.

Reviewer #2 (Remarks to the Author):

Very nice job responding to the reviewer comments

Response: Thanks for your positive comment.